# Chromosome-level genome and population genomics reveal evolutionary characteristics and conservation status of Chinese indigenous geese

Jing Ouyang[1,5], Sumei Zheng[1,5], Min Huang[2], Hongbo Tang[1], Xiaohui Qiu[3], Shoujin Chen[4], Zhangzhang Wang[4], Zhongdong Zhou[4], Yuren Gao[1], Yanpeng Xiong[1], Guohua Zeng[3], Jimin Huang[3], Jiugen He[3], Jun Ren[2], Hao Chen [1✉] & Xueming Yan[1✉]

Geese are herbivorous birds that play an essential role in the agricultural economy. We construct the chromosome-level genome of a Chinese indigenous goose (the Xingguo gray goose, XGG; *Anser cygnoides*) and analyze the adaptation of fat storage capacity in the goose liver during the evolution of *Anatidae*. Genomic resequencing of 994 geese is used to investigate the genetic relationships of geese, which supports the dual origin of geese (*Anser cygnoides* and *Anser anser*). Chinese indigenous geese show higher genetic diversity than European geese, and a scientific conservation program can be established to preserve genetic variation for each breed. We also find that a 14-bp insertion in endothelin receptor B subtype 2 (*EDNRB2*) that determines the white plumage of Chinese domestic geese is a natural mutation, and the linkaged alleles rapidly increase in frequency as a result of genetic hitchhiking, leading to the formation of completely different haplotypes of white geese under strong artificial selection. These genomic resources and our findings will facilitate marker-assisted breeding of geese and provide a foundation for further research on geese genetics and evolution.

[1] College of Life Science, Jiangxi Science and Technology Normal University, Nanchang 330013, China. [2] College of Animal Science, South China Agricultural University, Guangzhou 510642, China. [3] Animal Husbandry and Veterinary Bureau of Xingguo County, Guangzhou, Jiangxi Province, China. [4] Xingguo Grey Goose breeding Farm, Guangzhou, Jiangxi Province, China. [5] These authors contributed equally: Jing Ouyang, Sumei Zheng. ✉email: haochen-jxau@hotmail.com; xuemingyan@hotmail.com

The domestic goose is one of the most economically important agricultural animals, as it can provide nutritious meat, eggs, and fatty liver. The geese bones from Tianluoshan in the lower Yangtze River showed that Chinese geese were domesticated more than 7,000 years ago[1]. Under the influences of diverse economic cultures and geographical structures, 30 Chinese indigenous geese breeds with different phenotypic features and production performance have gradually been formed[2]. The genome assemblies of three domestic geese have been recently completed, but only the hybrid Tianfu goose (TFG) was assembled to the chromosome level[3]. Chinese indigenous geese (the Sichuan white goose [SCW] and the Zhedong white goose [ZDW]) assemblies are still at the scaffold level[4,5]. The lack of high-quality genomes of Chinese geese restricts their molecular genetic research and breeding practice. Recently, many studies combining genome assemblies and comparative genomics have explored the genetic evolution of birds[6,7]. However, the genetic characteristics of domestic geese in the *Anatidae* remain relatively unknown.

Chinese geese (except for the Yili goose breed) descended from the swan goose (*Anser cygnoides*, ACy), while European geese (e.g., the Landaise goose, LDG) descended from the greylag goose (*Anser anser*, AAn)[8]. Southern China is an advantageous production area for geese breeding due to its plump float grass (*Cyperaceae*). The Xingguo gray goose (XGG), Fengcheng gray goose (FCG), Guangfeng white goose (GFW), and Lianhua white goose (LHW) in Jiangxi Province as well as a Hunan breed (the Lingxian white goose [LXW]) are widely distributed in Jiangxi Province (Fig. 1) and are excellent Chinese indigenous breeds, generally characterized by strong disease resistance and superior meat taste[2]. However, only for XGG has a national conservation farm been established, and the other breeds are under provincial and municipal protection with poor population uniformity. Many breeds are at risk due to unclear pedigree structure and inbreeding depression[9,10]. Population structure is an important determinant of biodiversity evaluation and is the basis for the protection and utilization of genetic resources. Current studies on the genetic structure and genetic diversity of domestic geese are based on the mitochondrial genome, microsatellite markers, or genotyping by sequencing[8,11,12,13], while studies using whole-genome resequencing data are insufficient.

Feather color, which is one of the main features of domestic geese, is considered an economically important trait. White feathers are preferred in consumer products (e.g., mattresses and coats) and are preferred for meat production due to the faster growth rate of birds with white plumage. Chinese geese have only white and gray colors, so geese can generally be divided into white geese and gray geese. Previous reports revealed that the mutation of melanocyte-inducing transcription factor (*MITF*) and the haplotype differences of tyrosinase (*TYR*) may be related to the white feather trait of ZDW[14]. Xi et al. identified a 14-bp insertion in exon 3 of endothelin receptor B subtype 2 (*EDNRB2*) gene that was associated with white plumage in Gang geese[15]. Wen et al. suggested that an 18-bp deletion in the intron of the KIT protooncogene, receptor tyrosine kinase (*KIT*) gene influenced the white feather phenotype of Chinese geese[16]. Despite extensive research on the white feathers in domestic geese, its genetic basis has not been fully elucidated.

To provide a chromosome-level genome for Chinese indigenous geese, we used a hybrid *de novo* approach including PacBio, Illumina, 10× genomics, BioNano, and Hi-C technologies combined with comparative genome analysis to explore the biological characteristics of geese during the evolution of *Anatidae*. Large-scale resequencing of 994 geese was carried out for population genetic analysis to reveal the genetic diversity, genetic differentiation, and resource conservation status. Additionally, we used selective sweep and allele frequency differences to detect the causal mutations and origin of the plumage color of Chinese domestic geese, and the selection signatures of the XGG population were also explored. Our study not only provides invaluable data resources for global geese research but also contributes to germplasm resource exploration, the causal mutation of white plumage in Chinese domestic geese, and goose breeding.

## Results

**An improved Chinese indigenous goose genome.** The genome of a female XGG was sequenced and *de novo* assembled by PacBio, Illumina HiSeq, 10×genomics, BioNano optical genome mapping,

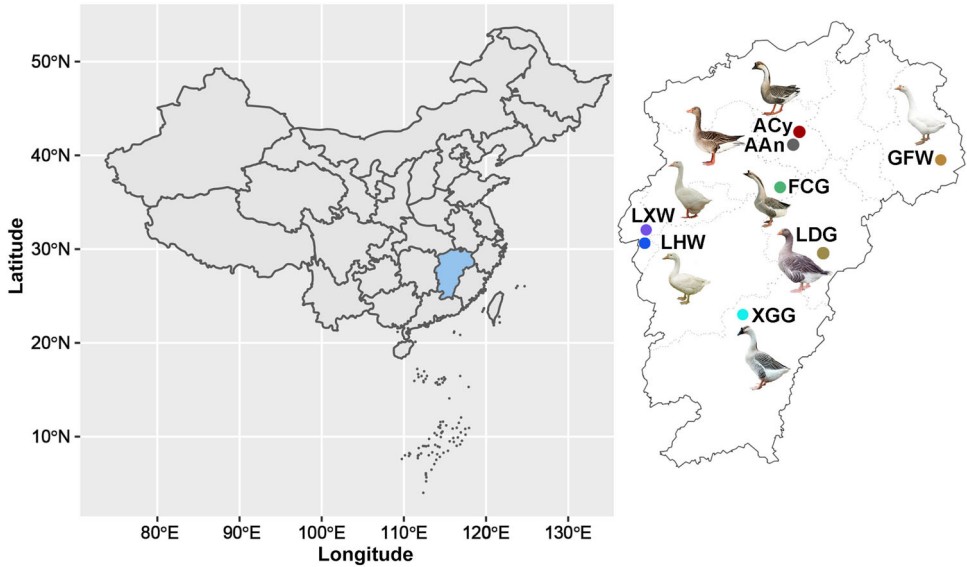

**Fig. 1 Morphology and geographical distribution of domestic geese breeds and their ancestors.** Circles are colored according to population. Domestic geese breeds including XGG (Xingguo gray goose), LXW (Lingxian white goose), LHW (Lianhua white goose), FCG (Fengcheng gray goose), and GFW (Guangfeng white goose) in China and LDG (Landaise goose) in Europe. Ancestors include ACy (swan goose, *Anser cygnoides*) and AAn (greylag goose, *Anser anser*).

**Table 1 Comparison of quality metrics within goose genome assemblies.**

| Feature | XGG | TFG[3] | ZDW[4] | SCW[5] |
|---|---|---|---|---|
| Assembly level | chromosome | chromosome | scaffold | scaffold |
| Genome coverage (×) | 562.4 | 324.63 | 107.35 | 56.2 |
| Genome size (bp) | 1,163,486,048 | 1,277,099,016 | 1,208,661,181 | 1,198,802,839 |
| Number of chromosomes | 39 | 39 | - | - |
| Number of contigs (>2 kb) | 2648 | 2771 | 60,979 | 53,336 |
| Number of scaffolds (>2 kb) | 2242 | 2055 | 1050 | 1837 |
| N50 contig length (bp) | 19,834,234 | 1,849,874 | 27,602 | 35,032 |
| N50 scaffold length (bp) | 77,964,326 | 33,116,532 | 5,202,740 | 5,103,766 |
| GC content | 42.07% | 42.15% | 38% | 41.68% |
| Repetitive sequences | 10.16% | 8.67% | 6.33% | 6.9% |
| Number of gene models | 17,448 | 17,568 | 16,150 | 16,288 |
| Number of exons | 170,177 | 152,392 | 158,713 | 167,532 |

*XGG* Xingguo gray goose, *TFG* Tianfu goose, *ZDW* Zhedong white goose, *SCW* Sichuan white goose.

and Hi-C sequencing technology, in total producing 686.12 Gb of sequences with ~562.4×genome coverage (Table 1, Supplementary Fig. 1, Supplementary Table 1, and Supplementary Method 1). The 123.62 Gb of sequences (~101.33×) obtained from the PacBio platform was used for initial contig assembly and then combined with the 10× genomics (120.90 Gb, ~99.10×) and BioNano (167.71 Gb, 137.47×) optical mapping technology to acquire highly continuous super-scaffolds. Next, the high-quality Illumina paired-end reads (136.14 Gb, ~111.59×) were used for error correction. The super-scaffolds were then improved to chromosome level using valid Hi-C data (137.75 Gb, ~112.91×). In total, Hi-C linking information supported 1.13 Gb (97.65%) of scaffold sequences being anchored, ordered, and oriented to 39 pseudo-chromosomes (Supplementary Fig. 2; Supplementary Tables 2–3). The final constructed genome contained 2,242 scaffolds with 1.16 Gb of sequences, a contig N50 length of 19.83 Mb, a scaffold N50 of 77.96 Mb, and a guanine-cytosine (GC) content of 42.07% with normal ratios of A, T, G, and C (Table 1 and Supplementary Table 4). Compared with other geese and chromosome-level bird genomes, XGG showed a large scaffold N50 (77.96 Mb) except for the kakapo[17] (*Strigops habroptila*) among 16 avian genomes (Supplementary Table 5) ranging from 3.89 Mb to 83.24 Mb (Fig. 2a). In particular, our constructed genome for XGG displayed 2.35-fold, 14.99-fold, and 15.28-fold improvements in scaffold contiguity over that of a currently reported hybrid goose (TFG)[3] and the other two Chinese indigenous geese breeds (SCW and ZDW)[4,5] (Table 1).

To further assess the accuracy of the scaffolded genome, we realigned the Illumina paired-end reads to the XGG genome with high mapping (99.19%) and high coverage rates (97.96%), generally reflecting the base accuracy of the reliable genome (Supplementary Table 6). The genome was further evaluated by CEGMA[18], and 92.3% of 248 core genes from six eukaryotic model organisms could be identified (Supplementary Table 7), a value that was significantly higher than the evaluation results (85.08%) of TFG[3]. We also used all three published geese genomes and 12 other chromosome-level bird genomes to carry out BUSCO[19] analyses to delve into the completeness of the XGG genome. The results showed that 95.7% of complete genes and 1.0% of fragments of genes were identified from the 8,338 core genes in the Aves dataset, better than the evaluation results of TFG[3] (Fig. 2a). In addition, we predicted 17,448 non-redundant protein-coding genes with an average of 9.52 exons per gene (Supplementary Fig. 3, Supplementary Method 2, and Supplementary Table 8). The number of genes in the XGG genome was close to that of TFG (Table 1), and 17,135 (98.2%) were functionally annotated (Supplementary Fig. 4 and Supplementary Table 9). For the repetitive annotation, the XGG genome

contained 10.17% non-redundant repeat sequences, including 2.04% tandem repeats and 8.55% transposable elements (Supplementary Tables 10–11). Although the repeat sequences would be folded in the assembly process, resulting in gaps and increasing the difficulty of assembly, we still managed to assemble more repeat sequences than TFG (8.67%), and thus may offer more genetic information (Table 1). We also identified 424 microRNAs (miRNAs), 371 transfer RNAs (tRNAs), 234 ribosomal RNAs (rRNAs), and 346 small nuclear RNAs (snRNAs) (Supplementary Table 12). The main features of the XGG genome are summarized in a Circos plot in Fig. 2b.

**Identification of sex chromosomes.** The sex determination system, which is of great evolutionary and ecological significance, is ZZ/ZW in birds. To further explore such genomic characteristics in XGG, we integrated three steps to accurately identify the sex chromosomes, something that has not been accomplished previously in geese genomes[3–5]. A total of 10 sex chromosomes from seven chromosome-level avian genomes in public databases were selected as reference chromosomes (Supplementary Table 13). After sequence splitting, homology alignment, and classification, we identified XGG's Z and W linked sequences containing 22 scaffolds (Hic_3 length 78,059,545 bp, total length 78,520,308 bp) and two scaffolds (Hic_4 length 18,190,528 bp, total length 18,196,581 bp), respectively (Supplementary Table 14). Also, the collinearity to the chromosome level showed that Hic_3 had a closer synteny with the Z chromosome of the Pekin duck (GCF_009819795.1), while Hic_4 presented a closer synteny to the W chromosome of Pekin duck (Supplementary Fig. 5). The average sequencing depth of autosomes and sex chromosomes of 103 females and 59 males used in the following population structure analyses (Supplementary Table 15) also supported the identification results (Supplementary Method 3). These findings suggested that Hic_3 and Hic_4 were most likely the Z and W chromosomes of XGG, respectively.

**Phylogenetic tree and gene family evolution.** To explore the evolutionary relationships and establish the phylogenetic position of the Chinese goose (*Anser cygnoides*) in the *Anatidae* clade, we chose 10 representative birds from seven genera in *Anatidae*, with *Phasianidae* as an outgroup for phylogenetic analysis (Supplementary Table 16). We identified 16,055 gene families across 12 avian species, and 6,371 single-copy genes were employed to construct a phylogenetic tree. As shown in the time-calibrated phylogeny (Fig. 2c), *Anser* (e.g., XGG) and *Cygnus* (e.g., Black swan) had the closest genetic relationship, with a divergence time around 19.6 Mya. The goose and the pink-footed goose as sister

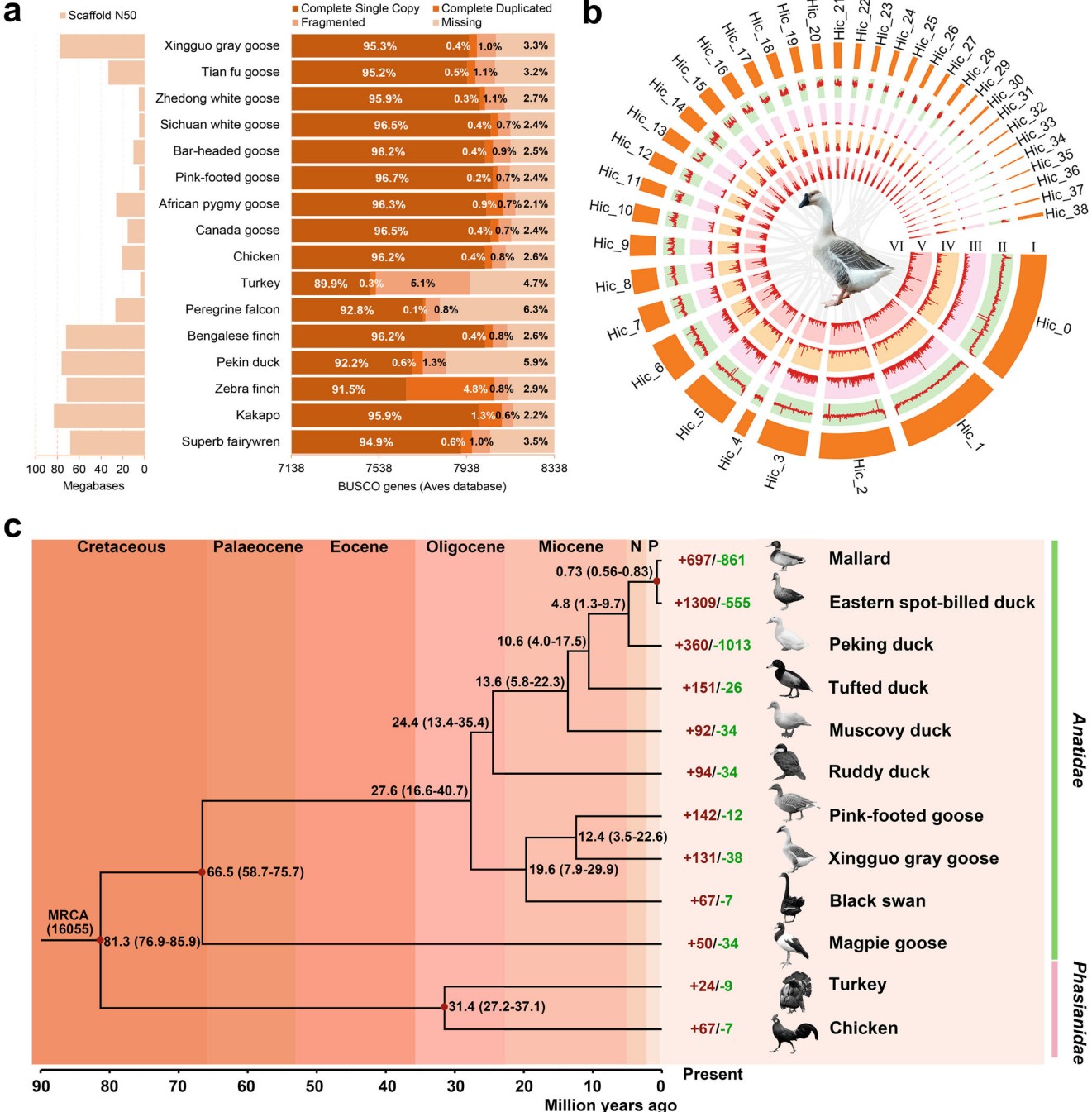

**Fig. 2 Quality assessment of XGG genome and comparative genomic analysis. a** Quality assessment of avian genomes. The left is the size of scaffold N50 and the right is the results of the evaluation using BUSCO Genes of Aves Database. **b** Genome landscape of the XGG. I, Chromosome number; II, GC density in 100-kb sliding windows; III, Repeat density; IV, Gene density of positive-strand (+); V, Gene density of negative strand (−); VI, Paralogous relationships in XGG chromosomes. **c** Phylogenetic tree and gene family expansion-and-contraction analysis. The red and green numbers represent the significantly expanded and contracted gene families, respectively. Divergence time was estimated based on four calibration points (red circles). N Pliocene, P Pleistocene, MRCA most recent common ancestor.

branches derived from the same ancestor diverged between 3.5 and 22.6 Mya.

A comparison of four geese genomes (XGG, SCW, ZDW, and TFG) showed that a total of 11,733 gene families, while 9,390 gene families were shared. We detected 11,648 gene families in XGG, of which 15 XGG-specific gene families included 38 genes (Supplementary Fig. 6). Such lineage-specific gene families were mainly enriched in "dynactin complex" (GO:0005869, $P = 3.53 \times 10^{-13}$), and "actin filament" (GO:0005884,

$P = 1.28 \times 10^{-10}$; Supplementary Table 17). We also observed 131 significantly expanded gene families (268 genes) in the goose lineage (Fig. 2c). The most significantly enriched GO term was "transmembrane-ephrin receptor activity" (GO:0005005, $P = 1.80 \times 10^{-16}$; Supplementary Table 18), comprising nine genes of the *EPH* gene family that play critical roles in neuronal network formation, hearing, and the olfactory system[20–22].

Furthermore, we used the branch-site model and likelihood ratio tests to identify the genes in the goose that have evolved under

positive selection. A total of 1,136 genes (Supplementary Data 1) appeared to be under positive selection, some of which were involved in lipid metabolism, such as "ATP-dependent activity (GO:0140657, $P = 2.97 \times 10^{-10}$)" "lipid transport" (GO:0006869, $P = 7.77 \times 10^{-9}$), "phospholipid binding" (GO:0005543, $P = 7.68 \times 10^{-8}$), "cholesterol metabolism" (hsa04979, $P = 6.89 \times 10^{-5}$), and "Fat digestion and absorption" (hsa04975, $P = 4.34 \times 10^{-3}$, Supplementary Data 2). Several positively selected genes (PSGs) in the enrichment terms such as *APOB* (Apolipoprotein B)[23] and *MTTP* (Microsomal Triglyceride Transfer Protein)[24] have been reported to be associated with liver steatosis in geese. Additionally, some PSGs were also prominently enriched in RNA-related processes (GO:0034660 and hsa03018; Supplementary Data 2).

**A high-quality genome-wide variation dataset from 845 geese.** A total of 994 geese from China and Europe were selected for whole-genome resequencing. The Chinese group consisted of 772 XGG, 51 FCG, 50 GFW, 11 LHW, 50 LXW, and 5 ACy, and the European group consisted of 50 LDG and 5 AAn (Supplementary Data 3). We used the Illumina platform to sequence 772 XGG with an average depth of 1×, while the remaining 222 geese (51 FCG, 50 GFW, 11 LHW, 50 LXW, 5 ACy, 5 AAn, and 50 LDG) were sequenced with an average depth of 10×. These sequencing reads were aligned with the reference genome XGG assembled above, 772 XGG (1×) yielded 12,415,004 SNPs, while 222 geese (10×) yielded 13,008,900 SNPs that were more abundant than XGG population, largely due to breed diversity and higher sequencing depth (Supplementary Method 4). The genetic relationship between each pair of individuals was calculated for every population and a phylogenetic tree was constructed to identify the repeated or closely related samples. From both analyses, those with lower sequencing quality of paired individuals that shared a proportion of identity-by-descent (IBD) >50% were removed. In total, we deleted 136 individuals of XGG and 13 individuals of other breeds (Supplementary Data 3). The 11,029,910 SNPs with minor allele frequency (MAF) >1% and call rate >90% of 845 geese (including 636 XGG, 50 FCG, 46 GFW, 9 LHW, 45 LXW, 49 LDG, 5 ACy, and 5 AAn) were used for subsequent analysis (https://bigd.big.ac.cn/gvm/). Among these SNPs, we detected 6,462,809 (58.59%) intergenic and 4,567,101 (41.41%) genic mutations, including 318,809 (2.89%) intergenic SNPs that were located within 1-kb up and downstream of the gene, 52,301 (0.47%) missense mutations, and 443 splicing mutations as possibly important in geese genetic diversity.

**The population genetic structure of Chinese indigenous geese.** A neighbor-joining (NJ) tree of the aforementioned 845 geese showed that Chinese and European geese were divided into two major branches (Supplementary Fig. 7). We observed an LHW and an AAn individual deviated from the population branch, while XGG was significantly distant from other populations (Supplementary Fig. 7). Considering the unbalanced sample size, we randomly selected 50 XGG to reduce the population structure deviation, and the outlier individuals from LHW and AAn were removed. Finally, 257 geese comprising 50 XGG, 50 FCG, 46 GFW, 8 LHW, 45 LXW, 49 LDG, 5 ACy, and 4 AAn were kept for subsequent population analyses (Fig. 3). Here, we noted that Chinese indigenous breeds (XGG, FCG, GFW, LHW, and LXW) and ACy had the closest genetic affinity, while LDG was close to AAn (Fig. 3a, b), thus supporting previous reports on the dual origin assumption of domestic geese[25]. The NJ tree indicated that all individuals were clustered together according to their breeds except for Chinese LHW and LXW, consistent with the clustering results of PCA (Fig. 3a, c). Seeking to clarify the genetic differentiation and admixed history of Chinese indigenous geese, we

further examined the genetic structures of these populations (Supplementary Fig. 8). As the lineage number *K* increased, XGG, GFW, and FCG separated as unique ancestral components corresponding to $K = 3$, $K = 4$, and $K = 5$, respectively. Population structure analysis showed that cross-validation error was the lowest at $K = 5$ (Supplementary Fig. 8a), while LHW and LXW still represented the same lineage (Fig. 3d), implying that LHW and LXW may be the same breed.

**Analysis of the genetic diversity of Chinese indigenous geese.** To further reflect the real population structure of Chinese indigenous geese, we performed genetic diversity analyses on 990 geese (we removed four duplicate individuals of XGG) (Table 2 and Supplementary Figs. 9–12). Statistical results showed that the number of SNPs (NSNP) in the Chinese group (4,685,880–5,245,562) was clearly higher than that in the European group (828,686–1,703,829). The observed heterozygosity (Ho), expected heterozygosity (He) and inbreeding coefficient (F) had the same trend as those for NSNP, indicating that Chinese geese had higher genetic diversity than European geese. The results for NSNP, F, $F_{ROH}$, and linkage disequilibrium (LD) showed that the genetic diversity of ACy was slightly higher than that of Chinese breeds. Among these Chinese indigenous geese, we found that XGG appeared to have the highest level of genetic diversity, as reflected in having the most SNPs (4,996,204), the largest Ho (0.28), and He (0.28), the smallest F (0.06), and a larger genetic distance (DST = 0.24). Notably, LHW had the least number of SNPs (4,695,880), the highest inbreeding coefficient ($F = 0.23$ and $F_{ROH} = 16.30\%$), the smallest DST (0.04), and the largest LD decay distance ($r^2_{(0.3)} = 1.63$ kb), suggesting the lowest genetic diversity of LHW.

**Selection signatures between Chinese white and gray goose populations.** We estimated zFst between white geese (46 GFW, 8 LHW, and 45 LXW) and gray geese (50 FCG and 50 XGG) and conducted zHp analysis on the white population to scan for genomic signatures of selection. A total of 13 overlapping genes were detected, with *EDNRB2* and *POLR1D* (RNA Polymerase I And III Subunit D) on chromosome 15 having the most significant zFst (11.23) and zHp (−3.84) values (Fig. 4a, b and Supplementary Table 19). We then analyzed the haplotypes of the significantly differentiated region *EDNRB2-POLR1D* in 672 individuals from 16 populations (Supplementary Table 20). Unexpectedly, a specific cluster of haplotypes (including 285 variants) was discovered here that could distinctly separate white and gray geese (Fig. 4c, d and Supplementary Fig. 13). Allele frequency analysis showed that a total of 25 SNPs and 2 InDels tended to be fixed in white geese, and the allele frequency difference of these mutations in white and gray geese was more than 0.9 (Supplementary Table 21). Further functional annotation found that a 14-bp frameshift insertion occurred in the coding region of *EDNRB2*. This insertion, absent in ancestral populations and closely related species, was the only derived allele among all 285 variants. The frequency was as low as 1.89% in gray geese (FCG), while being completely fixed in white geese (Supplementary Tables 23–24), consistent with the conclusion that white geese were artificially bred from a few gray geese after the mutation[26]. In addition, the haplotype (including 14-bp) of FCG was closest to the haplotypes of white geese (53 nucleotide differences), and thus it could be reasonably speculated that the key mutation (14-bp) affecting the white feathers of Chinese geese may have originated from FCG (the breed name may not have appeared at that time) or other populations not studied here.

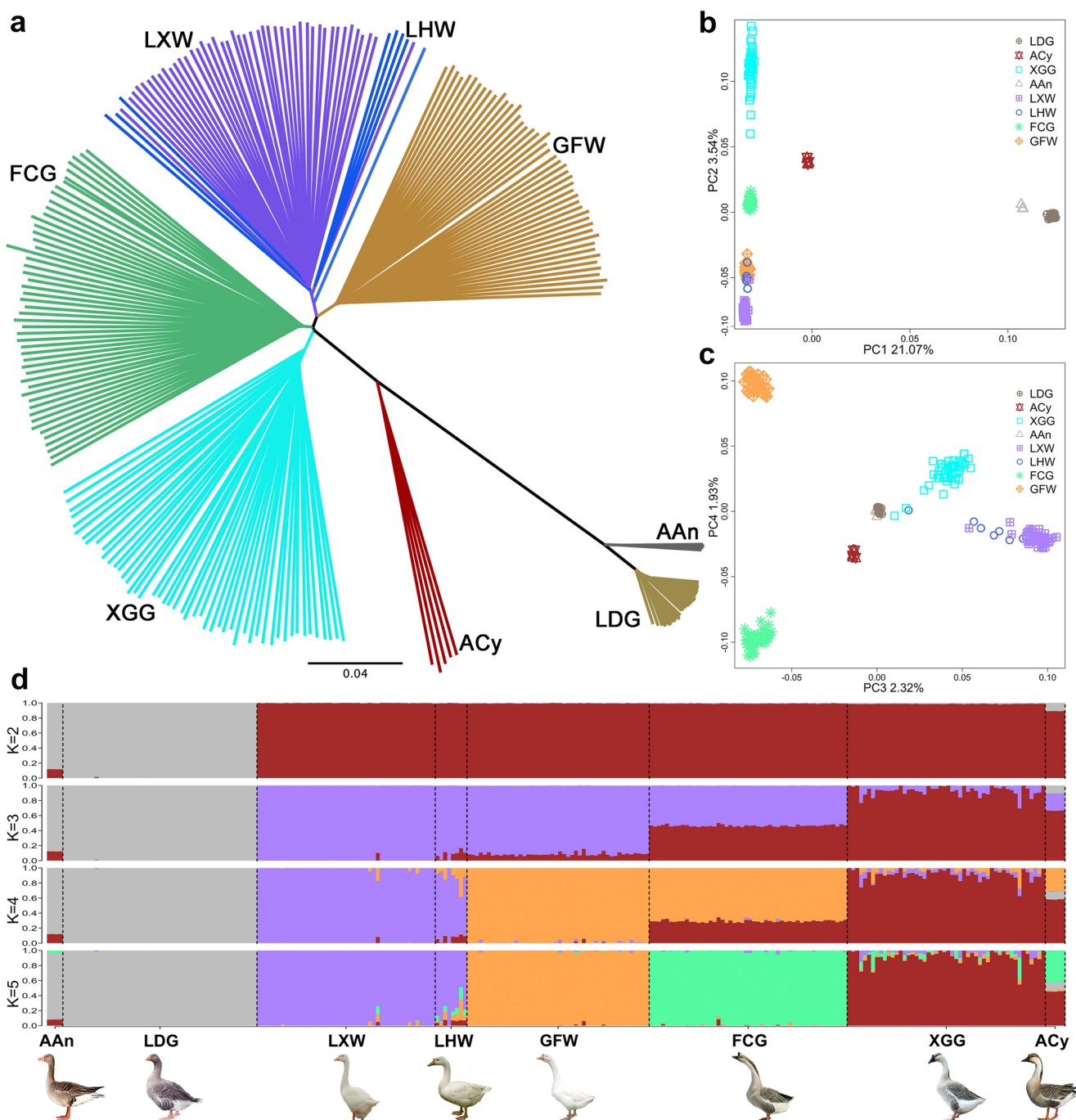

**Fig. 3 Phylogenetic relationships and population structure of Chinese indigenous geese breeds. a** The NJ tree was constructed by identical-by-state matrix of 257 goose individuals. **b**, **c** Principal component analysis of 257 goose individuals. PC1, PC2, PC3, and PC4 explained 21.07%, 3.54%, 2.32%, and 1.93% of the observed variance, respectively. **d** Ancestral composition of Chinese geese breeds. $K = 2$–5 represented the number of assumed ancestors ($K$), and each color represents an ancestral lineage.

**Selection signatures in the XGG population.** A total of 200 XGG individuals were randomly chosen for zFst statistics with 149 other Chinese geese (50 FCG, 46 GFW, 45 LXW, and 8 LHW), and the heterozygosity (zHp) analysis was conducted in all 636 XGG to detect the genomic regions under selection. We further focused on 21 overlapped functional genes based on zFst and zHp analyses (Supplementary Table 22); some immune-related genes such as *DAB2* (DAB Adaptor Protein 2), have been reported to be one of the differentially expressed gene markers for sheep mastitis resistance[27], and the difference in expression was directly related to disease severity[28]. *MYO1F* (Myosin IF) is mainly expressed in

neutrophils, and molecular experiments in mice have demonstrated that this gene may help neutrophils fight infection and that it plays a critical role in acute and chronic inflammatory diseases[29,30]. In actual observation, we found that epidermal cysts (Supplementary Fig. 14) are slowly growing benign cysts that commonly occurred on the feet geese in the XGG population, but not in other geese breeds. Immunity is the organism's own defense mechanism, and the onset of some diseases is immune-related[31]. We inferred that these immune-related genes may be involved in the occurrence of epidermal cysts in the feet of XGG. In addition, we found that the gene prolactin receptor (*PRLR*)

**Table 2 Genetic diversity among eight geese populations.**

| Breed | Number | NSNP | Ho | He | F | DST | $F_{ROH}$(%) | $R^2_{(0.3)}$(kb) |
|---|---|---|---|---|---|---|---|---|
| ACy | 5 | 5,245,562 | 0.24 | 0.24 | 0.03 ± 0.04 | 0.23 ± 0.01 | 6.20 ± 4.24 | 0.77 |
| XGG | 768 | 4,996,204 | 0.28 | 0.28 | 0.06 ± 0.04 | 0.24 ± 0.01 | 10.99 ± 3.07 | 1.45 |
| FCG | 51 | 4,904,489 | 0.25 | 0.27 | 0.09 ± 0.04 | 0.23 ± 0.01 | 10.27 ± 2.69 | 0.84 |
| GFW | 50 | 4,815,415 | 0.25 | 0.26 | 0.11 ± 0.03 | 0.23 ± 0.01 | 10.09 ± 2.62 | 1.21 |
| LHW | 11 | 4,685,880 | 0.22 | 0.26 | 0.23 ± 0.18 | 0.04 ± 0.02 | 16.30 ± 16.6 | 1.63 |
| LXW | 50 | 4,708,718 | 0.24 | 0.26 | 0.13 ± 0.03 | 0.25 ± 0.01 | 10.09 ± 2.67 | 1.33 |
| AAn | 5 | 1,703,829 | 0.11 | 0.10 | 0.52 ± 0.35 | 0.09 ± 0.04 | 8.43 ± 4.44 | 1.89 |
| LDG | 50 | 828,686 | 0.05 | 0.05 | 0.78 ± 0.02 | 0.16 ± 0.01 | 26.88 ± 2.37 | 7.01 |

ACy *Anser cygnoides*; XGG Xingguo gray goose, FCG Fengcheng gray goose, GFW Guangfeng white goose, LHW Lianhua white goose, LXW Lingxian white goose, AAn *Anser anser*, LDG Landaise goose, NSNP the number of polymorphic snp, Ho observed heterozygosity, He expected heterozygosity, F inbreeding coefficient, DST intraspecific genetic distance, $F_{ROH}$ the proportion of homozygous fragments, $r^2_{(0.3)}$, the decay distance when the linkage disequilibrium is 0.3; ×value indicates standard errors.

was strongly selected in XGG population (Supplementary Table 22). This gene is a candidate genetic marker for reproductive traits and is considered to be a major gene influencing age at the first egg laying of chickens[32]. Meanwhile, *PRLR* could affect the egg production of ducks[33] and geese[34]. According to historical documents[2], XGG was committed to breeding in the two directions of growth rate and egg yield. We considered that *PRLR* has a certain promotion effect on the egg production of XGG, and breeding in the direction of egg production might have had some rewards during the long breeding process. Interestingly, Hic_asm_9.361 (annotated as *CLCA1* in chickens) and cysteine-rich hydrophobic domain 1 (*CHIC1*) were found to overlap 1136 PSGs mentioned in the above positive selection analysis. Although the functions of these genes have not been elucidated in geese, we speculated that they might play an important role in the adaptive evolution and breeding process of XGG; this could be the focus of future research.

## Discussion

XGG is an excellent goose breed, and it plays an essential role in the regional economy and conservation of the genetic resources of indigenous breeds. Here we report a chromosome-level reference genome of XGG and identified the sex chromosomes, thus providing valuable resources for the genetic investigation of Chinese indigenous geese. Further comparative genomic analysis indicated that the *EPH* gene family, which plays an important role in neuronal network formation[35], hearing[36], and olfactory systems[22], was significantly expanded in the goose lineage. This may be closely related to the high vigilance and stress response of geese. In addition, the olfactory system aids in bird foraging and can be used for migratory direction identification[37]. Although the goose used in this study has been domesticated by humans for thousands of years, it is essentially the descendant of a migratory bird (ACy). Therefore, the *EPH* gene family could help explain the migratory habits of geese as migratory birds. Geese have an excellent capacity to deposit fat in the liver. Positive selection analysis identified a series of PSGs enriched in lipid metabolism and RNA processing. Involved genes such as *APOB* participate in de novo synthesis of fatty acids and were associated with goose hepatic steatosis[38]. Cholesteryl ester transfer protein (*CETP*) has been shown to play a role in liver lipid metabolism[39]. Translocator protein (*TSPO*) regulates steatosis in nonalcoholic fatty liver disease[40]. There are also many potentially functionally related genes, such as scavenger receptor class B member 1 (*SCARB1*), *MTTP*, and lecithin-cholesterol acyltransferase (*LCAT*). In addition, Lu et al. found that microRNAs regulate multiple lipid synthesis and transport genes that are closely related to lipid metabolism in the goose liver[4]. These results suggested that fat deposition in the goose liver is a complex process regulated by a variety of signaling molecules and their pathways. These findings

help explain the genetic basis of fatty liver in domestic geese and provide insights into the genetic improvement of geese. Although this study demonstrated that these genes were critical factors for evolutionary adaptation in domestic geese, clarifying the specific functions related to these pathways will require further studies of their expression patterns and possible roles in growth and development.

We further resequenced the whole genomes of 994 wild and domestic geese to explore the population genetic structure, revealing two major clades of Chinese and European populations that supported the hypothesis that domestic geese have dual origins at the genome level[8]. Unexpectedly, the Chinese LXW and LHW were confused with each other and indistinguishable. A careful review of the relevant literature indicated that the central producing area of LXW was located in Lingxian County, Hunan province, but breeding also occurred in Lianhua County, Jiangxi province[2]. The origin and central areas of LHW were in Lianhua County, Jiangxi province, while neighboring areas such as Chaling and Youxian County of Hunan province were also represented. Given that LHW and LXW have very similar morphological characteristics and geographical distributions, we inferred that they were likely admixed at an early date due to human activities and were then bred locally. Genetic diversity analysis revealed that European geese, especially LDG, had extremely low levels of genetic diversity, a result that was consistent with previous findings[13,41]. As commercial geese breeds are dedicated to fattening the liver, the characteristics tend to be fixed, and exceptionally low genetic diversity reflects strong selection pressure. Most domestic geese showed lower genetic diversity than their ancestors (ACy and AAn), especially European domestic geese, probably due to genetic drift during population bottlenecks of initial domestication and subsequent artificial breeding[42]. Here, the genetic diversity of Chinese indigenous geese seemed to be at fairly healthy levels, except LHW which had the lowest genetic diversity, presumably due to multiple generations of inbreeding. Among those breeds, only XGG that has been conserved in a national conservation farm displayed the highest genetic diversity, a pattern that confirmed the uneven conservation of goose genetic resources in China and the necessity for scientific and effective preservation of genetic variation and rejuvenation programs.

Exploring the genetic mechanism of feather color has always been an interesting and exciting direction in the field of animal research. Our findings and those of previous studies[15] suggest that a 14-bp insertion in exon 3 of *EDNRB2* is a key mutation responsible for white feathers in Chinese geese. The insertion should be a recessive mutation in light of the extremely low frequency of heterozygotes in gray geese and fixation in white geese that experienced strong artificial selection. Chinese gray geese have plumage colors similar to those of their ancestors.

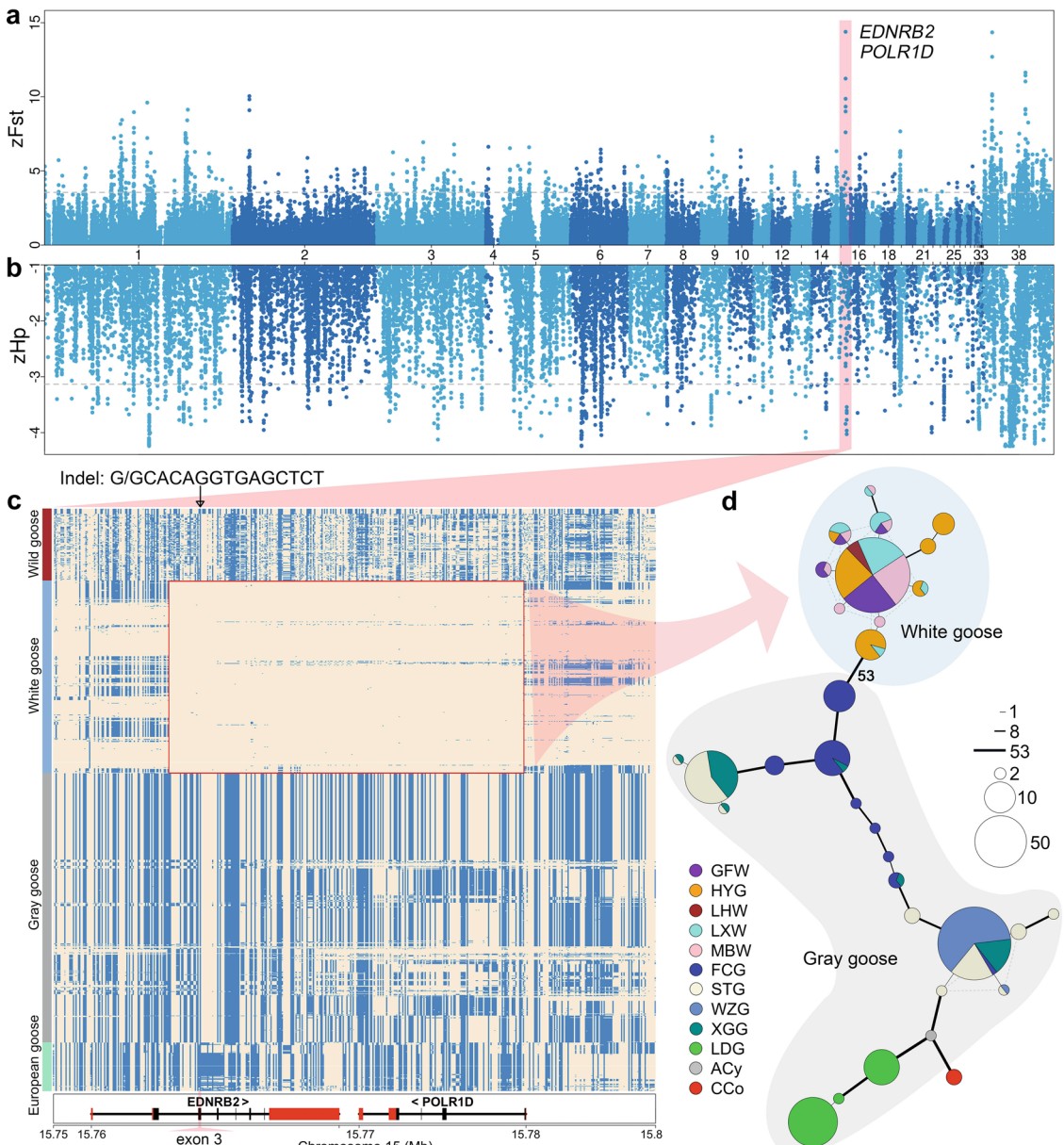

**Fig. 4 Selective signals for the white plumage phenotype of geese. a** Manhattan plot of zFst between white and gray geese. The *x*-axis of the Manhattan plot shows the ordered chromosome that is defined in Supplementary Table 2, and 38 represents Z chromosome. **b** Manhattan plot of zHp in white geese, with the positions matching zFst. The gray dashed line represents the top 1% cutoff. **c** The plot of the haplotype structure of variants around the *EDNRB2* and *POLR1D* genes in all domestic geese and wild populations (The genera *Anser* and *Cygnus* in the *Anatidae* family). Major and minor alleles in GFW are indicated by beige and light blue, respectively. The red box represents the unique haplotypes of white geese. The black arrow indicates the position (15,764,637 bp) of the candidate causal 14-bp insertion for the white geese. The red and black rectangles in the bottom box represent the UTRs and CDSs, respectively. **d** Haplotype network based on 285 SNPs and Indels from the *EDNRB2* gene (15,763,328 bp) to *POLR1D* gene (15,779,122 bp) on chromosome 15. Each circle represents a haplotype, and the size of the circle is proportional to the haplotype frequency. The line width and length represent the difference between haplotypes. GFW Guangfeng white goose, HYG Huoyan goose, LHW Lianhua white goose, LXW Lingxian white goose, MBW Mingbei white goose, FCG Fengcheng gray goose, STG Shitou goose, WZG Wuzong goose, XGG Xingguo gray goose, LDG Landaise goose, ACy *Anser cygnoides*, CCo *Cygnus columbianus*.

According to *Chinese Waterfowl*, gray geese were domesticated from wild ancestors, while white geese were artificially bred from a few gray geese after the mutation occurred[26]. However, the genetic diversity of white geese was not significantly lower than that of gray geese, indicating that white geese are ancient domesticated breeds; this can be verified by numerous Chinese poems describing white geese during the Tang Dynasty (618–907). In the haplotypes of white geese, only the 14-bp insertion was derived and fixed, and we thus considered that the

14-bp insertion was a natural mutation that occurred during the domestication process of the gray goose, and then these almost completely distinguished haplotypes were formed due to genetic hitchhiking under strong artificial selection. We also noted that the four individuals of FCG were 14-bp heterozygous carriers, indicating that the current FCG population is impure and may produce offspring with white feathers, although the proportion of 14-bp insertions is small. Additionally, agouti signaling protein (*ASIP*), OCA2 melanosomal transmembrane protein (*OCA2*),

*TYR*, tyrosinase-related protein 1 (*TYRP1*), melanocortin 1 receptor (*MC1R*), *MITF*, and *KIT* have also been studied for plumage coloration[14,16,43], but we have not identified these genes and potentially associated loci. In particular, Wen et al. revealed that an 18-bp deletion in the *KIT* gene was a key mutation in the white plumage of Chinese geese based on a total of 35 individuals from four breeds[16]. However, we did not find that the 18-bp deletion and possible SNPs, InDels, or CNVs located within the *KIT* gene significantly affected the feather color of geese, and the haplotype of the *KIT* gene did not show significant genetic differentiation between the white and gray geese (Supplementary Fig. 15). This reflected the importance of extensive and comprehensive sampling in population genetic research. In summary, our research has provided important resources for uncovering the evolutionary adaptations of domestic geese, and the results will further facilitate the breeding process of Chinese indigenous geese.

## Methods

**Sample collection**. All procedures involving animals used in this study have complied with guidelines for the care and utility of experimental animals established by the Ministry of Agriculture of China. The ethics committee of Jiangxi Science & Technology Normal University approved this study. We collected 994 blood samples (Supplementary Data 3) from four Jiangxi indigenous geese breeds (772 XGG, 51 FCG, 50 GFW, and 11 LHW), one Hunan breed (50 LXW), one European breed (50 LDG), and two wild populations (5 ACy and 5 AAn) in accordance with the principles and standards of animal welfare ethics. Meanwhile, three liver and muscle tissue samples were obtained from female XGG to aid in the genome assembly annotation process.

**DNA and RNA extraction**. The genomic DNA was extracted from blood samples using the traditional phenol-chloroform protocol (https://geneticeducation.co.in/phenol-chloroform-dna-extraction-basics-preparation-of-chemicals-and-protocol). We chose an adult female XGG collected from the national Xingguo Grey Goose Reserve in Jiangxi Province, China for *de novo* assembly. Furthermore, to assist genome annotation, total RNAs were extracted from different tissues (liver and muscle) according to the TRIzol (Invitogen) manufacturer's protocol for transcriptome sequencing.

**A chromosome-level genome assembly**. Genome assembly for XGG used a hybrid *de novo* assembly approach (Supplementary Fig. 1 and Supplementary Method 1). Initially, The PacBio subreads were assembled into contigs using wtdbg v2.4[44]. These contigs were then connected with Super-scaffolds with the linked-reads generated by the 10× Genomics Chromium via fragScaff v140324.1[45]. Gaps in the 10× Genomics assembly version were filled with the BioNano data using BioNano Solve v3.3[46]. This version was then polished iteratively two to three times to improve the single-base correction rate using Pilon v1.23[47] based on Illumina paired-end reads. Combining the scaffolds produced from the previous step with valid Hi-C data, we used LACHESIS v201701[48] *de novo* assembly pipeline to produce chromosome-level sequences. Finally, the consistency of the reconstructed sequences was comprehensively determined based on the extent to which the sequences covered the genome, and the integrity of the results was assessed by CEGMA v2.5[18]. The contiguity of the genome assembly was also compared to other geese and chromosome-level bird genomes (Supplementary Table 5) using BUSCO v5.2.2[19] analyses which searches the assembly for 8,338 universal single-copy orthologs from the Aves (odb10) database. Details regarding genome assembly were presented in Supplementary Method 1.

**Genome annotation**. Repeat sequence annotation in XGG genome was detected by combining homologous-based and *de novo* predictions (Supplementary Fig. 3). The repeat sequence library generated from *de novo* prediction and the homologous repeat sequence database Repbase[49] were integrated to screen for repetitive sequences by RepeatMasker v4.07[50] and in-house scripts (RepeatProteinMask). Then, Gene identification and functional annotation were performed via homology-based identification, *de novo* prediction, and transcriptome data-based approach. Protein sequences from homologous species (*Anas platyrhynchos*, *Anser cygnoides*, *Gallus gallus*, *Meleagris gallopavo*, and *Coturnix japonica*) were aligned to XGG genome using BLAST v2.2.28[51], and GeneWise v2.4.1[52] was then used to identify the gene structure. Augustus v3.3.3[53], Geneid v1.4[54], Genescan v3.1.2[55], GlimmerHMM v3.04[56], and SNAP v2013.11.29[57] were used for *de novo* prediction. To aid gene prediction, tissues from the liver and muscle retrieved from XGG were used to construct a normalized cDNA library. Transcriptome sequencing was performed on Illumina NOVASEQ 6000 platform. Transcriptome reads were aligned to XGG genome and using TopHat v2.0.8[58] to identify exons region, splice positions, and utilize Cufflinks v2.1.1[59] for transcript assembly. A total of 17,448

non-redundant genes were obtained using EVidenceModeler v1.1.1 and PASA v2.4.1[60]. Finally, all identified proteins were aligned to public databases such as SwissProt[61], NCBI nr[62], Pfam[63], KEGG[64], and InterPro[65] for functional annotation (Supplementary Fig. 4). Details regarding genome annotation were presented in Supplementary Method 2.

**Sex chromosomes recognition and assessment**. The avian sex chromosome sequences were downloaded from NCBI database to be used as reference sequences. These included six Z chromosomes and four W chromosomes from seven chromosome-level avian species (Supplementary Table 13). We split the genomic sequence of XGG into short reads and aligned them to reference sequences. The SAMtools v1.10[66] was then used to extract reads of Mapping Quality >30 for screening and classification. We discarded reads that aligned on different chromosomes or multi-reads aligned on the same chromosome and kept all results with the same mapping values or one result with the peak score. Finally, the scaffolds of sex chromosomes were classified according to the proportion of aligned reads and the degree of difference with the length of the reference sequences (Supplementary Method 3). For accuracy assessment, we initially used TBtools v1.06[67] to perform chromosome-level homology alignments between Pekin duck[68] and XGG. Afterward, the average sequencing depths of autosomes and sex chromosomes (W and Z) were calculated using SAMtools[66] (option: -depth) based on the resequencing data of 162 geese (50 LXW, 51 FCG, 50 GFW, and 11 LHW) with known sexes in subsequent population genetic analysis (Supplementary Table 15). We considered that the average sequencing depth of W chromosomes in males was ~0, while that of Z chromosomes was approximately the same as for autosomes, and the depth of Z and W chromosomes in females was about one-half that of autosomes.

**XGG-specific gene family identification**. To infer XGG-specific gene families in domestic geese, protein sequences of TFG, ZDW, and SCW (Supplementary Table 16) were downloaded from NCBI (https://www.ncbi.nlm.nih.gov/) and Ensembl (https://asia.ensembl.org/). We kept the longest transcript for each gene of four geese and used OrthoFinder v2.4.0[69] based on a Markov clustering algorithm to identify orthologous gene families with an E-value cutoff of 1e-5. The species-specific gene families were determined according to the presence or absence of genes for a given species.

**Phylogenetic tree reconstruction**. In addition to XGG, we also selected nine representative bird species of *Anatidae*, and reconstructed the *Anatidae* phylogeny with *Phasianidae* (chicken and turkey) as outgroups (Supplementary Table 16). Protein sequences of 6,371 single-copy genes from 12 species were generated using OrthoFinder and initially aligned by MAFFT v7.407[70] with default parameters; poorly aligned regions were then discarded by trimAl v1.4[71] based on a heuristic approach (option: -automated1). The resulting alignments of each gene family were concatenated into a super-alignment matrix using two popular software programs to reconstruct maximum likelihood (ML) trees: RAxML v8.2.12[72] with the PROTGAMMALGX model and IQ-TREE v2.1.1[73] with the self-estimated optimal substitution model. The topological structures generated by these two programs were similar.

**Species divergence time estimation**. The divergence times were estimated using MCMCTREE processed in the PAML v4.9j[74]. The Markov chain Monte Carlo (MCMC) analysis was run for 2,000,000 generations, with a sample frequency of 10 after a burn-in of 400,000 iterations. Meanwhile, four calibration times obtained from the TimeTree database (http://www.timetree.org/) were set for dating analysis: (a) *Gallus gallus–Anas zonorhyncha*: 77.0–86.0 Mya; (b) *Anseranas semipalmata–Anas zonorhyncha*: 59.0–76.0 Mya; (c) *Gallus gallus–Meleagris gallopavo*: 27.5–37.5 Mya; (d) *Anas platyrhynchos–Anas zonorhyncha*: 0.27–0.79 Mya. Then, we used Tracer v1.7.1[75] to check the convergence of the chains to a stationary distribution.

**Expansion and contraction of gene families**. To estimate the changes in gene repertoire in the XGG, 16,055 orthologous families identified by OrthoFinder from the 12 species described above were used for expansion and contraction analysis. Among them, we applied a random birth-death model of CAFÉ v4.2.1[76] for inference. The phylogenetic tree topology and branch lengths were considered to infer the significance of changes in gene family size in each branch. The *P*-values of each lineage were calculated, and values <0.05 were considered significant.

**Positive selection analysis**. To detect PSGs in Chinese goose, the protein sequences and coding sequences (CDS) of all 6,371 single-copy genes were aligned using MAFFT, followed by pal2nal.pl v14[77] to generate codon alignments in PAML format. Finally, a gene tree was constructed for each single-copy gene using IQ-TREE, and the codeml program in PAML with the branch-site model was used to discover positive selection in particular lineages, where XGG was defined as the foreground branch and other birds as the background. The compared likelihood ratio tests of Model A (model = 2, NSsites = 2, $\omega > 1$) and a null hypothesis

(model = 2, NSsites = 2, $\omega = 1$) were analyzed by Chi-square tests, with $P < 0.05$ considered as significant.

**Whole-genome resequencing, SNP calling, and genotype imputation.** All 994 DNA samples were processed in whole-genome sequencing using the Illumina PE150 platform with an average insert size and read length of 350 bp and 150 bp, respectively. Two different genome-wide resequencing strategies were applied on 994 samples, 772 of which were XGG with an average sequencing depth of 1× and 222 other geese (51 FCG, 50 GFW, 50 LXW, 50 LDG, 11 LHW, 5 ACy, and 5 AAn) with an average sequencing depth of 10×. These genome sequencing data were aligned to the XGG reference, and variants were detected with Sentieon v201711.03[78] (twenty times faster than GATK) DNAseq pipeline following the best practices algorithms of GATK on a Tianhe-2 Supercomputer. For 772 XGG, we further used an R package "STITCH v1.68"[79] to impute to the whole-genome level. After quality control and filtering (Supplementary Method 4), 12,415,004 SNPs (MAF >1% and call rate > 90%) were finally detected in XGG population, and 13,008,900 SNPs (MAF >5%, call rate >90%) were identified in 222 geese. After filtering repeats or closely related individuals, 11,029,910 SNPs of 845 geese with MAF >1% and call rate >90% merged from the datasets of 636 XGG, and 209 other geese were used for subsequent analysis. The final SNP datasets were further analyzed and classified by SnpEff v5.0[80] according to the gene annotation of the XGG reference genome.

**Population genetic structure analyses.** We utilized the PLINK v1.9[81] -genome option to calculate the genetic relationship (shared proportion of IBD) between pairs of individuals for each breed and removed the one with a lower sequencing detection rate for PI_HAT > 0.5 pairs (Supplementary Data 3). For the remaining 636 XGG, a total of 50 XGG were randomly selected to form 257 individual datasets (10,767,528 SNPs) with other breeds for population genetic analysis. Then, an NJ tree of 257 individuals was reconstructed using PHYLIP v3.69[82] based on the identical-by-state matrix calculated by the "plink -distance-matrix" command and finally visualized via Figtree v1.4.2 (http://tree.bio.ed.ac.uk/software/figtree/). Afterward, we ran GCTA v1.92[83] using the above SNPs to produce a genetic relationship matrix and extract the first four eigenvectors for PCA analysis. SNP sites with $r^2 > 0.2$ were eliminated to decrease the influence of LD (measured as $r^2$), and 749,370 SNPs were retained to estimate the ancestral consanguinity component (from $K = 2$ to 10) of each population by ADMIXTURE v1.3.0[84] with 10-fold cross-validation.

**Genetic diversity analysis.** We removed possible duplicate individuals from the sampling process to estimate the genetic diversity. For the remaining 990 individuals, we calculated seven parameters, comprising the number of SNPs (NSNP), observed heterozygosity (Ho), expected heterozygosity (He), inbreeding coefficient (F), intraspecific genetic distance (DST), runs of homozygosity (ROH)-based inbreeding coefficient ($F_{ROH}$), and linkage disequilibrium (LD). Ho and He were estimated with the option -hardy in PLINK. F was calculated by the "plink -het" command at each site. The command "plink -distance square 1-ibs" was used to measure the DST. The PLINK parameters -homozyg -homozyg-snp 30 -homozyg-kb 30 was set to compute $F_{ROH}$ with the genome size of 1.17 Gb. The square of the correlation coefficient ($r^2$) was used to measure the degree of linkage between each pair of SNP alleles; $r^2$ was estimated by the parameters -r2-ld-window-kb 500-ld-window-r2 0 in PLINK.

**Selection signatures between Chinese white and gray geese.** To uncover the hidden genomic region of artificial selection signatures for Chinese white geese, we used 10-kb sliding windows for selective sweeps based on whole-genome sequencing data, calculating Fst between 99 white geese (46 GFW, 8 LHW, and 45 LXW) and 100 gray geese (50 FCG and 50 XGG) and performing Hp analysis on white geese. To characterize the germplasm characteristics of XGG, we randomly selected 200 individuals of XGG and 149 other Chinese geese (50 FCG, 46 GFW, 45 LXW, and 8 LHW) for Fst analysis, and analyzed the heterozygosity (Hp) of XGG. The average Fst and Hp values of each window were calculated by filtering the windows with the number of SNPs <10. Fst and Hp values were normalized (zFst and zHp) by subtracting genome-wide mean and dividing by the standard deviation, and in-house R scripts were used for visualization. The top 1% was set as the significance threshold, and genes from the overlapping region of zFst and zHp were considered convincing functional candidates.

**Haplotype analysis around EDNRB2-POLR1D region.** To further investigate the EDNRB2-POLR1D gene region (Chr15: 15.7–15.8 Mb) associated with white plumage, the resequencing data of 672 individuals (the average depth >10×) from 16 populations (89 wild geese, 222 white domestic geese, and 361 gray domestic geese; Supplementary Table 20) in our own bird database was used for verification. We then constructed the heatmap of the haplotype around the EDNRB2-POLR1D region. The details were as follows: (i) First, we used XGG as a reference genome for mutation detection and obtained 521 SNPs and Indels with MAF >5% and call rate>90% for all 672 accessions. (ii) Then, we phased haplotypes for 521 SNPs and Indels using the fastPhase function with 1000 iterations in Beagle v5.1[85] and visualized via a haplotype heatmap constructed using an in-house R script. (iii)

Finally, we constructed a haplotype network based on 285 SNPs and Indels in the EDNRB2-POLR1D gene region using the "haploNet" command in the R package "pegas v1.1"[86], and the pairwise differences between haplotypes for 285 SNPs and Indels at the EDNRB2-POLR1D loci were calculated.

**Reporting summary.** Further information on research design is available in the Nature Research Reporting Summary linked to this article.

## Data availability

The Xingguo gray goose genome reported in this study has been deposited in the Genome Warehouse in BIG Data Center (https://bigd.big.ac.cn/gwh/) under accession number GWHBAAW00000000. The genome resequencing data for 994 domestic and wild geese have been deposited in the NCBI database as BioProject PRJNA678815. The SNP dataset of domestic and wild geese has been deposited in the Genome Variation Map in BIG Data Center (https://bigd.big.ac.cn/gvm/) under accession number GVM000131. The numerical source data for graphs have been deposited in Figshare (https://doi.org/10.6084/m9.figshare.20929474).

## Code availability

Analytical pipelines and code are available on Zenodo (doi:10.5281/zenodo.6613753).

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

## Acknowledgements

This study is supported by the National Natural Science Foundation of China (no. 31860622 and 32060735) and the Technology Research Project of Jiangxi Provincial Education Department (no. GJJ180595).

## Author contributions

X.Y., H.C., and J.R. conceived and designed the study. H.C., J.O., and S.Z. analyzed the data. H.C., S.Z., and X.Y. wrote the paper. H.C., S.Z., M.H., and H.T. performed the bioinformatics analyses. S.Z., H.T. Y.G., and Y.P. collected data and performed sequencing. X.Q. S.C. Z.W. Z.Z. G.Z. J.M.H., and J.G.H. collected data. The author(s) read and approved the final manuscript.

## Ethics approval and consent to participate

All procedures used for this study and involved birds complied with guidelines for the care and utility of experimental animals established by the Ministry of Agriculture of China. The ethics committee of Jiangxi Science and Technology Normal University approved this study.

## Competing interests

The authors declare no competing interests.
