## [Peer Review File · Communications Biology]

Reviewers' comments:

Reviewer #1 (Remarks to the Author):

This paper explores a consider number of goose resequencing data, comparative genomics and evolutionary analysis. However, these lots of analyses focuses too much information which hinders the manuscript cannot answer one question in more detail. Analyses cannot fully match the aims of this study.

The first aim of this study is to study the phylogenetic position and evolutionary history of domestic goose. However, the background didn't give the sufficient reason for this aim. Previous studies explore the goose genome and comparative studies (Lu et al, 2013). Current study should explain why and how to improve the domestication events of goose. The results also focus the novel findings and genetic relationship of aisa goose and geese in other countries.

The 2,3 and 4 aims focused on the genetic structure of Jiangxi geese, and conservation strategy. But the results described too much un-related information about the conservation strategy, i.e., sex chromosomes, genome assembly. The positive selection events in the XGG didn't answer the local adaptability of the XGG and environmental relationship.

In terms of the conservation issue, authors stated "we then developed a selection principle in whihc recurrent cross-breeding of ganders among families and female geese were equally reserved for breeding offspring..." (line 315-320). These conclusions didn't provide the novel information, and also didn't do simulation of conversion, population genetic analysis for this so-called plan.

Lots of sequenced data are useful for bird biology and conservation study. However, too much aims of this study have been stated, and some conclusions are not accurate.

As stated above, this study did lots of work, but should focuse on genome assembly improvement, comparative genomics and genomic structure of local geese. Or shorten these parts of analyses, then focus on the conservation strategy adjustment based on the molecular evidences.

Reviewer #2 (Remarks to the Author):

This manuscript provides an in-depth work characterizing the geese species and breeds from a genomics perspective either in terms of comparative analyses as either in population genomics. These approaches reveal to be a novel and very comprehensive genomics characterization which unravels many important aspects in geese. For instance, the first goose genome that is not a hybrid; the sex chromosomes identified for the first time; among others.

There is quite a lot of work done and I even thought it could be partitioned into several articles that could also provide the opportunity to improve and deepen the specific results. In fact, this amount of work makes it difficult to understand the flow or how the analyses are serving the mentioned objectives or motivation (and is also harder for the author's to organize so many things in a single manuscript).

Generally I found that the text is well-written and it is visible that the authors have put a great effort to make it clear. However, I present some minor suggestions below to help improve clarity, line-by-line, that should be checked (whenever possible) throughout the main text, figures and supplementary material.

The presented methods and results are generally well conceived and presented, however, they need textual improvements for clarity and to ensure the reader is not lost by the lack of information or other aspects.

1) I feel that the manuscript title doesn't entirely make justice to the kind of work developed

throughout the manuscript. Here, if possible, I would suggest to change to "Historical relatedness, conservation status, comparative and population genomics in the Chinese indigenous geese" (note "geese" here) or any alternative form which shows the most important genomics work that has been accomplished here, where (as a main part) and as stated in Discussion and other sections is "the first chromosome-level" of this breed and among all sequenced geese to date.

2) The Abstract section could also be improved to better describe the manuscript contents, maybe if the author's rewrite it following the organization into Background, Results and Conclusion subsections it may result in a better abstract (without necessarily containing these titles).

3) In the Background section, I feel that a small paragraph by the end, would greatly help to quickly understand the manuscript layout, specifically the Methods and Results sections. This paragraph would be showing what work is being presented, or how the objectives are being addressed by the several analyses, for instance, "In section X, we present these analyses (1,2,3...) to investigate this idea and/or to meet mentioned objective 1; second, we perform these analyses (1,2,3...) to investigate idea 2 and/or to meet mentioned objective 2; etc."

4) As the Methods section is placed at the end of the manuscript it makes difficult to assess the Results (reading directly from Background) taking into account the different breeds, populations, species involved, and numbers used (e.g., 994, 990, 845, 257, 149, etc), so I believe that improvements to fill these gaps could help in terms of clarity and readability. This section should focus on describing the analyses, any problems faced and solutions. It should be clear how analyses are performed (software used, etc) and how data (types), numbers, etc are attained. Additionally, sometimes I felt that Methods or Results subsections could benefit some re-organization, as for instance, the positive selection analyses are not following the phylogeny analyses section, which breaks the necessary flow.

5) The terms "assembly", "chromosome-level" and "scaffolding" cause some confusion (See also comment in line 547 below). Each term pertains to a different step in establishing the genome contiguity. "Assembly" and similar, should be used to only refer to the process of reaching the contigs sequences from the sequencing reads. "Scaffolding" and similar, should be used for the process of reaching scaffolds sequences from the contigs sequences. And "chromosome-level" should be used when the scaffolds give rise to "chromosome-level" sequences like those in the Human genome. By reading the text (lines 99-156, and others) it becomes difficult to understand at which point is the reconstruction process of the genome or what is the term referring to. I understand that "assembly" is generally used to refer to the process of piecing together (and not necessarily the stage of reconstruction), but when there are other stages of reconstruction, it becomes confusing. If the authors write "assembly" instead of a synonym, it is perceived as only referring to this delimited step and any related analyses, but should not refer to others; and so forth. For instance, Lines 118-119, should the "assembled genome of XGG showed the largest scaffold" be modified into "scaffolded genome of XGG showed the largest scaffold" ; Line 121: "our newly assembled genome" modified into "our newly scaffolded genome"? (For reference: <https://doi.org/10.1371/journal.pcbi.1006994>)

Detailed comments:

Line 21: In Discussion it is mentioned that this chromosome-level genome is constructed for the first time, why not here?

Line 42: "charactics" is correct? Could "features" be a better word?

Lines 48, 94: replace "goose" with "geese"?

Lines 50-55: Following readability and sections organization, I would suggest that the various geese

breeds and any other species from the study should be similarly and briefly described here (framing them also in terms of their original species/populations and domestic/wild; perhaps with the help of an additional main table) in addition to the current ones and any value for this study, like why are they important? It could be possible to move some information from Methods-Experimental animals section to this paragraph. Furthermore, they are also mentioned in Abstract, hence why not here?

Line 75: five genes are here described and associated to plumage coloration in geese, however, there is a study, that mentions other genes being studied regarding plumage coloration (<https://doi.org/10.1186/s12864-015-1924-3>). It appears from the whole article, that only EDNRB2 has been studied and found to be partly causal for coloration (lines 357-358). From these, TYR is common, and others are not mentioned. Has any in this set of 7 genes been found and could these genes (differences) also significantly influence plumage coloration?

Line 79: is not clear whether this refers to the same previously mentioned breeds or if these references refers to different ones.

Line 89: what kind of "relationship"?

Line 91: "origination" should read "origin"? Or another better word.

Line 108: the term "reduplication" is not clear, please explain. What is its importance for the scaffolding process?

Line 112: what is "mount rate"? Do the authors mean coverage breadth (<https://doi.org/10.1038/nrg3642>, <https://doi.org/10.1371/journal.pcbi.1006994>)? In fact, coverage and read depth can be used to mean the same, but it would make more sense to distinguish both by using different and more appropriate terms with sequencing depth applied to refer to (e.g., ~99,10x) cases and coverage applied to the percentage that refers to the amount of sequencing that spans the entire genome.

Lines 117-120: please provide references for these studies, regarding mentioned "genomes of other geese", "studies in birds" and the "16 avian draft genomes".

Line 119: kakapo bird, is not clear if is a goose or any other species. Should the scientific/latin name be provided? References?

Line 121: "of the typical Chinese local goose" refers to the previously mentioned "female XGG"? Please clarify.

Lines 123-124: references needed.

Line 125: replace "genome assembly" with "scaffolded genome"?

Lines 126, 128: "the XGG genome draft" the word 'draft' should not be used since the genome is in scaffold level? The terms "draft genome" are usually employed to mean low accuracy and/or initial reconstruction. See also other occurrences.

Line 129: reference missing for CEGMA [?].

Line 130: "covered" means "identified" and/or "annotated"?

Line 103, 134 and others: Supplementary material indication sometimes is given with "Supplementary" word and in others is not.

Line 132: First occurrence of "Tian fu goose". It seems the only one in text and in figures that has no abbreviation, suggesting "TFG". Please modify accordingly. Additionally, I suggest the use of the (existing) abbreviations should be much more common in text, figures, tables, etc to quickly identify which breed or species is being referred (e.g., Line 151: "TFG" could be used instead or in addition to "hybrid goose").

Line 138: missing reference for the mentioned "results of Tian fu goose".

Line 140-142 needs rephrasing, it starts by giving a value associated to exons, and next values are suddenly given in tandem. It should be easier to read in "value item," form.

Lines 143-144: "draft genome" and "chromosome-level goose genome", which is which? Is "draft" appropriate term?

Line 149: "folded" is the correct term? What it means?

Line 161: "three methods" which are? Or should it be "one method consisting of 3 steps?" Replace "locate" with "identify"?

Line 162: "of the typical Chinese indigenous goose" could be removed? Could the whole sentence be reduced to "To further explore such genomic characteristics in XGG, we integrated three steps to accurately identify the sex chromosomes, which has not been accomplished in hybrid goose genome [15]." Feel free to improve.

Line 164: "panel" word could be replaced by "reference" or "control"?

Line 166: delete first "and" word. Are these the previously mentioned "three methods"?

Line 172: a word is missing: "closer"? Also from the Peking duck?

What is the correct spelling of Peking Duck with "g" or without? It appears to be misspelled in several occurrences, whereas in Fig 1d appears with "g" but not in Fig 1a and in the main text (Line 171, etc).

Lines 172-176: the involvement of 103 females and 59 males is not previously explained, how this happened and why? Which breeds?

Line 186: replace "applied" with "employed"?

Lines 196, 200, 202: which "goose and chicken"? Plural?

Line 204: "four goose genomes" which are, please specify? Would "geese" be more correct?

Line 205: provide figure identification.

Line 206: "and" should read "of which"? Is it 11,648+15?

Line 206: "containing" replaced by "including"?

Line 207: "gene" word is missing.

Line 210: the figure 1d shows 131 expanded and 38 contracted gene families, is there also any idea of which are the 38 gene families contracted in XGG (plus number of genes) and possible effects?

Line 222: The id GO:0051186 is shown as obsolete, is there any alternative GO?

Line 226: missing citation for GO.

Line 226-234: Please clarify this sentence. Are you referring to aforementioned GO terms or genes? Is FAS another GO id? Is this related to the positive selection detected?

Line 236: What proportion or number of individuals pertains to each population/breed?

Line 239: the 772 are XGG and the 222 are? "1X" and "10X" should specify their meaning.

Line 246: "Eventually" should be replaced or removed. 845 geese are all XGG?

Line 248: the number of SNPs can be removed, just say "Among these SNPs"

Line 254: the 845 individuals are the same as the 845 from previous section?
"neighbor-joining (NJ) tree" is abbreviated here for the first time, but only used once (line 703) throughout the text. It is recommended the use distinguishing abbreviations for different NJ trees / methods.

Line 264-265: It becomes unclear if the new dataset of 257 was important for the next paragraph or if this still results from previous analyses. From the previous two sentences it is not clear how the 257 dataset was attained and what breeds are included. I would suggest to start the following sentence (l.265) with "Here," instead of "Similarly,".

Line 281: why not 845 or even 257 individuals instead of 990? Were these four individuals also removed from previous analyses?

Lines 283 and 294: it seems both LGD and LHW had the "lowest genetic diversity"? Please clarify.

Lines 306-307: which one is this NJ tree, the Fig. S7 or Fig. 2B?

Line 333: Table S24 shows gene EDNRB, but is this different from EDBRB2?

Line 345: "presented" or "present"?

Lines 378-380: I am not sure if "usually" is the right word. Diseases can also be onset from several non-immune related genetic defects (affecting DNA, RNAs, proteins, etc). Cysts are often benign or noncancerous, hence to write that they may be related to immune genes, requires evidence. Is there any study or analysis that can confirm or point to the association of these cysts with immune-related genes?

Lines 390-393: To identify the function, have the authors performed homology searches of this gene in databases, for instance in NCBI? Do the results confirm the speculation?

Line 423 : the LXW first occurrence is in line 267 (no previous or current abbreviation), but the indigenous geese from Jiangxi province were described in lines 50-55. Should the LXW also be included here?

Line 425: "chronicles" could mean "literature"?

Line 425-430: following the previous point (line 423), should this (and other) sentences be moved (or reused) to Background section near the mentioned paragraph? See also comments in lines 50-55.

Lines 497-500: this sentence should be rephrased. Did you mean: added by accident?

Lines 515-517: "and the data fill gaps in our knowledge and facilitate further" could be replaced by "and the results further facilitate"?

Lines 532-533: why this separation of 994 into 772 and 222? What took to these numbers? (See also line 651)

Line 535: this adult female is the same individual as the one mentioned in line 530? Should this description be moved to the above section (or Background)? See also comments in lines 50-55.

Lines 538-545: should these values be repeated (here and in Results)?

Line 539-541: how were the Super-scaffolds reconstructed? Which software (no citation)? Wtdbg2 version missing.

Line 541: the "preliminary assembly" refers to the PacBio assembly or to the resulting scaffolds, upon applying the linked-reads from 10x platform?

Line 542: the "assembled version" refers to the scaffolds upon being filled by Bionano data?

Lines 541-542: how were the gaps filled? Which software (no citation)?

Line 544: Pilon version missing.

Line 545: what was combined with Hi-C data?

Line 546: LACHESIS version missing.

Line 547: "chromosome-level scaffolds" are these sequences chromosomes (like those in the Human genome) or still scaffolds? In case these are not exactly chromosomes, I would suggest to start refer to them as either scaffolds (or super-scaffolds) or eventually "near chromosome-level sequences". In supplementary material the term "pseudo-chromosome" is used, is this with same meaning?

Line 548: "assembled" could be replaced by "reconstructed" or "scaffolded"?

Line 549: Was CEGMA used for all this step or in addition to...? Please improve. Software version missing.

Line 559: Rebase needs a citation.

Line 560: RepeatMasker version missing.

Line 561 and others: Careful should be taken when writing "gene prediction" or "prediction" as opposed to "gene identification". Gene predictions focus on detecting novel gene structures or models and any regulatory regions, which tend to have not previously been detected (E.g., GenScan (<https://doi.org/10.1006/jmbi.1997.0951>), Genefinder, Genewise (<https://doi.org/10.1101/gr.1865504>)). Whereas gene identification would be a better term for simply identifying genes that are à priori known in other species, using strategies like those employing BLAST and any of the above.

Line 565: which was the software used, BLAST or BLAT? If the former, then the reference needs to be adjusted accordingly. Could <https://doi.org/10.1186/1471-2105-10-421> this be appropriate? Software version missing.

Line 566: "genewise" would be better written as "GeneWise". Version missing.

Lines 566-568: GeneWise is used to predict gene structure and the following three are used for "de novo prediction.". Are all four employed for the same purpose? GlimmerHMM version missing.

Lines 568-570: this sentence needs rephrasing. "genes were annotated by the prediction results", how and why? Functional annotation? Why "combined with RNA-seq comparison data"? Did this help to identify genes or extra data was used in the process? Which "comparison data" was used? Software version missing.

Line 571: "NR" should be better written as "NCBI nr "? Citations are missing in all cases.

Line 576: replace "for reference, including" with "to be used as reference sequences. These included"? From which databases were the sequences downloaded? Provide citations and any sequence IDs or accessions in Table S13.

Line 579: "the" is missing. SAMtools version missing.

Lines 583-584: "the scaffolds" repeats.

Line 586-587: why TBtools is not described in the detailed Method S3? Version is missing. Suggest moving "for accuracy assessment, " to the start of the sentence.

Line 588-589: how were the depths calculated? Which software (no citation)?

Line 596: which protein sequences were used? All? Did this included the identified sex chromosomes?

Lines 600-602: Did the clusters help to determine the gene families? How?

Line 604: why 10 species?

Line 607: what does "self-blast" mean? Was BLAST used? Was BLAST also used in above subsection?

Line 618: PAML version missing.

Line 629: How was the number of 16,055 orthologous gene families reached?

Lines 635-638: this seems to be the same methodology used in phylogenetics analyses, could this be resumed to a simplified reference to the section and the resulting phylogeny and alignments? Were the concatenated MSA and tree used in studying positive selection?

Line 639: "EMBOS" is better written as "EMBOSS".

Line 640: "paml" is better written as "PAML".

Lines 638-646: The positive selection analyses should be performed for each DNA gene and to this end, each MSA and corresponding phylogenetic tree, should be estimated based on the same orthologous sequence dataset. Despite the important back-translation process, can the authors ensure the whole procedure is correct? Due to the degeneracy of genetic code, is it ensured that the original codons are being used for both cases? Or, instead could you download and use the (protein-corresponding) original DNA sequences? Why wasn't a DNA-based phylogenetic tree estimated for each gene?

Additionally, the study (<https://doi.org/10.1093/molbev/msq303>) mentions that these types of

analyses are suitable for detecting episodic positive selection which affects only a subset of codons, why the authors opted for branch-site models and not other, for instance, the site-models? Would different models results alter the manuscript conclusions relative to this part? Here I would also suggest the authors to consider the read and/or use of LMAP and LMAP_S software published in BMC Bioinformatics.

Line 648: replace "implemented" with "processed" or other better word.

Line 653: Sentieon version missing?

Lines 651, 659, 660: should indicate what the (1X, 10X) are.

Lines 651, 659: I find it odd that the authors have divided into a dataset of 772 XGG only and 222 of diverse breeds. Moreover, they are sequenced at different depths, why? Could the highest number of detected SNPs be due to the highest breed diversity in 222 dataset?

Line 664: SnpEff version missing.

Line 667: --genome is an option not a command, unless "plink --genome" is the complete command. Hereafter, the software options and commands start to appear in the main text. They should all be discriminated in full for all cases and analyses and not just for a few. Hence, I suggest to include (and move) all of them (present and omitted) in Methods supplementary materials files in the appropriate sections. This way also avoiding to increase the manuscript length. This should also include software with graphical interfaces, by indicating the all main functions used.

Line 675: "phylip" should be written "PHYLIP".

Line 676: Figtree is missing citation (or as URL).

Line 682: ADMIXTURE has a version missing?

Lines 693, 697: no need to repeat software versions, here and elsewhere.

Line 701: This sentence could be reduced to refer to previous section.

Line 703: "phylip" should be written "PHYLIP" and version number at first occurrence, unless different versions were used.

Lines 713-714: why these numbers?

Line 718: "language" can be deleted. Alternatively, a citation should be used for R.

Line 736: "pegas" should have a version number?

Lines 738-747: are these related to section starting in line 709? Why are they separate?

Line 741-742: suggest moving " to characterize the germplasm characteristics of XGG", to the beginning of sentence?

Figures:

Overall figures are good, but captions should be checked accordingly to previous detailed comments. Figure 2a could benefit from arranging the geese figures next to the points marked on the map (thus

removing the doubled abbreviations). As it is, does not look like a legend. Or, perhaps the legends from a) and e) could be unified into a single legend. Even though, Figure 2a would make more sense as an initial figure to show the breeds being described (i.e., since Background lines 50-55).

Tables:

Overall tables are good, but legends and titles should be checked accordingly to previous detailed comments.

Supplementary Material:

It is visible that many of the referenced software and other material is not referenced in main text, which also makes hard to fully understand the Methods section. Ideally, they all should at minimum be enumerated, or at least those that are not at the detail level.

Line 42: hybridScaffold requires citation.

Line 50: BWA requires citation.

Lines 53-79: In Stage 5, it is confusing because the type of data from Hi-C technology is not mentioned and the stage 4 data supposedly in scaffold form is not mentioned. Thus taking to the idea that the data being used is from previous stages, is this correct?

Line 92: the study mentions the capture of material for transcriptome (main text, lines 530-531; 569) and here the "assembled transcriptome" is mentioned (and further in lines 127, 138), but so far no section has described these methods? Are they related?

Line 108: TRF (Tandem Repeats Finder) should be unabbreviated and cited.

Line 118: uclust needs a citation.

Line 130: BLAST could have a different citation (please see above), or in addition to this.

Line 143: EVM and PASA are the same citation, but should also be cited next to PASA.

Line 147: BLASTp should be cited. Should use <https://doi.org/10.1093/nar/25.17.3389>

Lines 148-150: missing citations. See also main text line 571.

Line 155: missing citation for GO.

Line 161: "blast" is an abbreviation and should be written as BLAST. This should be enforced for all occurrences (even to file formats), i.e., correct spelling.

Line 181: How was quality control achieved? Used any software?

Line 203: which "above reference genome"? Please specify.

Line 204: "sam" file format is an abbreviation.

Line 205: "bam" file format is an abbreviation.

Line 206: PCR unabbreviation required. Where did this come from?

Line 207, 209: "Haplotyper" and "GVCfTyper" are software or functions of Sentieon? Citations might be missing. "gvcf" is an abbreviation.

Line 209: bcftools requires citation. Should be written as "BCFtools".

Lines 211-213, 219-220: What is the meaning of "|"? Should be translated accordingly. What is the meaning of these metrics? A list of abbreviations could help establish the meanings for each abbreviation in main text and related files.

Lines 213, 217, 220: "maf" is Minor Allele Frequency (MAF)?

Table S1:

It seems peculiar that only two sequencing technologies have insert sizes specified.

In Line 6, it is mentioned a 20 Kb insert, but is missing in table. Are others missing too? And read-length?

Table S3:

What is the difference between both (bp) values?

Table S4:
Table title, "Base" could be modified to "Nucleotide".

Table S8:
It is confusing to have one column with both species and software. The column on the right also needs better title. Software should be cited.

Table S9:
Needs improved titles in columns.

Table S10:
Needs improved title in "Type" column, like "Software" or "Method".

Table S12:
Needs improved titles in columns.

Table S14:
Needs improved title.

Table S15: as mentioned in the Note, this table is showing sequencing depth for each individual and their chromosomes. Title can be improved and Note could be deleted.

Table S18, S19 and S20:
"P value" should be written as p-value, here and elsewhere.

Table S19:
Gene Symbols should also be upper-case as done with Table S21.

Table S22:
Needs improved titles in 3 columns.

Table S25:
Allele1 and Allele2: is one considered the reference allele? Which one?
NCHROBS title could be clearer.
Line 393: spelling.

Figure S3:
This figure mentions Trinity software, but not found in any text. Was it used?

Reviewer #3 (Remarks to the Author):

Ouyang et al. have constructed a chromosome-level reference genome of Chinese domestic goose of swan goose origin, performed large-scale re-sequencing of Chinese domestic geese and examined signs of selection and population structure within the Chinese domestic goose breeds. In addition, Ouyang et al. studied phylogenetic relationship of Anatidae species and suggested a conservation strategy for indigenous Chinese domestic goose breeds. The authors identified signatures of selection between white and grey Chinese domestic geese and examined further one of these identified candidate genes (EDNRB2), as this gene has an effect on coat and plumage color of several domestic animal species. The authors identified a 14-bp insertion in the EDNRB2 gene to be perfectly associated with the white plumage phenotype of Chinese domestic geese. The authors suspected an introgressed origin for the insertion region, but failed to pinpoint potential species from which this insertion could have originated. Ouyang et al. also identified selection signatures between the Xingguo gray goose

and other goose breeds, potentially conveying breed specific characteristics.

An annotated high-quality chromosome-level genome assembly of Chinese domestic goose is a welcome addition to the genomic resources of domestic geese. The sample size of genome-resequencing was impressive. It was also noteworthy that this research focused on indigenous breeds and conservation of their genetic resources, as often research on domestic animals tends to focus on high-output commercial breeds. Also, this study made interesting findings regarding the genomic background of the white plumage phenotype. This study is a valuable addition to goose domestication genetics. However, I think that the manuscript could be written more clearly, especially the Introduction section. At some parts the Introduction seemed to assume that the reader is familiar with the special characteristics of domestic geese, which might not be the case. Also, in the Result section some parts could be moved to Discussion section. The authors state that the 14-bp insertion must have introgressed from another unknown goose species, but fail to provide any tangible evidence for this. I do not think the authors can make such a definitive claim based on the evidence presented in the paper and should consider alternative explanations as well, e.g. hard selective sweep and genetic hitchhiking. More detailed comments to improve the paper are presented below line by line.

Detailed comments:

Title: The title "Historical relatedness, conservation status, and signatures of selection in the Chinese indigenous goose" seems misleading as there were no analyses studying historical relatedness. To me, historical relatedness implies a temporal aspect, e.g. samples of domestic goose from museum or archaeological contexts or simulations. Please re-phrase the title to better reflect the content of the current manuscript. E.g. chromosome assembly, genome annotation and re-sequencing would much better reflect the manuscript's content.

Abstract:

Line 20: In this line you mention conservation biology. Conservation biology refers to protection of endangered species and you should explain that you refer to conservation of indigenous domestic breeds and their genetic resources.

Line 23-24: You refer to fatty liver and unique immune system of the goose. However, the sentence requires the reader to know what fatty liver and unique immune system of goose are. Replace "fatty liver" with high fat storage capability in goose liver or something similar. Also very briefly explain, what you mean by unique immune system of the goose. How it is unique? Do you mean high disease tolerance?

Line 24-27: This sentence does not exactly reflect the content of this paper. You have not studied the Yili breed nor all the Chinese domestic goose breeds. This information about the Yili breed is established in previous studies, but your abstract gives the impression that this breed is involved in your study. I suggest that you rephrase this sentence as "Population structure analysis verified that all studied Chinese domestic goose breeds descended from the swan goose (*Anser cygnoides*) and the studied European domestic goose breed from the greylag goose (*Anser anser*)" or similar.

Line 31: Change "the 14-bp" to "a 14-bp"

Line 32-33: See my later comments in the main text regarding the interspecific hybridization and rephrase the text accordingly.

Line 34: Change "the marker-assisted" to "a marker-assisted"

Main text:

Line 39: Please add reference for this.

Line 39-41: Could you please state the evidence to which this is based on in the Bao 1996 paper. I don't have access to the original reference and it seems to be in Chinese, thus it would be beneficial for the reader if you could state to which archeological evidence this date is based on. E.g. context of the archeological finds, size of the bones etc.

Line 41-43: Please state what the distinctive characteristics of the goose are and what their economically important values are. The word "characteristics" is misspelled, please correct. Also, the goose is general term and refers to variety of species. Do you refer to all goose species or do you mean specifically domestic goose (and domestic goose generally or specifically Chinese domestic goose). As domestic geese has dual origin (swan goose and greylag goose) it is important that you establish which population you are referring to by domestic goose.

Line 48: Has all 30 breeds been characterized with genetic analyses? If this is not the case, change genetic to phenotypic.

Line 58-60: Please state in more detail what the biological and economic significance of the goose means.

Line 66: I think you need add a little bit background information why plumage color is of special interest. Now plumage color is suddenly brought up without any connection to what is previously said in the paper. Maybe it could be linked to color polymorphism segregating in domestic geese populations or early selection target in goose domestication and thus of special interest, for example.

Line 66-69: This sentence could be understood so that you are saying that microsatellites, mtDNA or GBS-data are not reliable, only genome re-sequencing is reliable. Please re-phrase. These methods are reliable, but can only be used to answer certain specific research questions or provide much more limited information. Also, I feel that you are not giving proper credit to genome-wide studies already done on Chinese domestic goose, for example:

- Deng et al. 2021 Integrative analysis of histomorphology, transcriptome and whole genome resequencing identified DIO2 gene as a crucial gene for the protuberant knob located on forehead in geese
- Li et al. 2020 Pacific Biosciences assembly with Hi-C mapping generates an improved, chromosome-level goose genome
- Liu et al. 2021 Genomic characteristics of four different geese populations in China
- Ren et al. 2021 Pooled Sequencing Analysis of Geese (*Anser cygnoides*) Reveals Genomic Variations Associated With Feather Color
- Wen et al. 2021 Genomic scan revealed KIT gene underlying white/gray plumage color in Chinese domestic geese
- Xi et al. 2020 A 14-bp insertion in endothelin receptor B-like (EDNRB2) is associated with white plumage in Chinese geese
- Gao et al. 2016 Genome and metagenome analyses reveal adaptive evolution of the host and interaction with the gut microbiota in the goose
- Lu et al. 2015 The goose genome sequence leads to insights into the evolution of waterfowl and susceptibility to fatty liver

Even though you mention many of these references later in the paper, here you give the impression that there is almost no genome-wide research on Chinese domestic goose, which is not the case. Referencing the previous studies places the results of your study in a correct context.

Line 76: Does ancestral populations mean the ancestral species the swan goose and the greylag goose?

Line 81-83: This sentence was not clear, application of what? Please rephrase.

Line 88: I'm a bit puzzled by the use of the word "germplasm" in here. Germplasm means: plant or

animal material (such as seeds, pollen, rootstock, or sperm) that is collected and stored chiefly for future use in breeding, conservation, or research (Merriam Webster dictionary). If I understood correctly your material was not germplasm as most breeds were not involved in conservation programs. I believe investigating signs of selection would better describe your analyses.

Line 91: Change "origination" to "origin"

Line 95-96: Change "the molecular mechanism of white feather formation" to "causal mutation of white plumage in Chinese domestic goose breeds"

Results section: Part of the results felt more discussion than results. There were also many references to other studies, but the Result section should be based on your results with minimal references to other studies. Please go through the Results section and move discussive parts to the Discussion section.

Line 102: Supplementary Fig. S1 has poor quality and thus difficult see the method overview. Please provide a higher quality image.

Line 110: Fig. 1b is before Fig.1a. Please make sure that figure panels are referenced in order of appearance, thus Fig.1a should come first.

Line 172-176: Is this based on re-sequencing data?

Line 176: Method S3 is mentioned before method S2. Please change the order

Line 187: Fig. 1d is before Fig. 1c

Line 237: Please include the species of the wild geese in parenthesis

Line 292-294: In the previous studies which you refer, the marker type was microsatellites. Microsatellites are multiallelic and highly polymorphic while SNPs are usually biallelic. Thus, due to inherent properties of these marker types, heterozygosity in SNPs is generally lower than in microsatellites. You cannot directly compare the heterozygosity value in microsatellites and SNPs. Please re-write this sentence.

Line 297-235: This whole chapter felt more Discussion than Results. Consider moving most of this chapter to Discussion.

Line 299-301: See the previous comment. The genetic diversity in Chinese domestic goose seemed to be on same level as the wild progenitor species the swan goose, with the exception of FROH. Compared to Landaise breed, the genetic diversity of Chinese domestic geese was much higher. Please also rephrase this sentence.
In addition, in line 278 you say that LHW and LXW might be same breed. Thus, establishing conservation strategy for LHW alone does not seem justified.

Line 306: Abbreviation IBS appears the first time, so please write in parenthesis what this abbreviation stands for.

Line 326-327: I suggest re-naming this paragraph as "Selections signatures between white and grey goose populations" to better reflect the analyses performed. To me this chapter gives the impression that you searched specifically the locus for white plumage and in this case GWAS would have been more appropriate. However, you performed first a sweep analysis to search for signatures of selection and then you focused on a particular candidate gene identified by this analysis because of its association with plumage and coat color in other species, or did I misunderstood the analyses you did?

Line 332: Should it be "genomic signatures of selection". Also, what do mean by shared genes? Do you mean that these genes were outliers both in zHp and zFst methods?

Line 337: Does ancestral populations mean swan and greylag goose? Are wild species and domestic geese listed in Table S26?

Line 353: Change "nucleotide difference by more than 53" to "more than 53 nucleotide differences"

Line 354-357: Could provide more information on this. E.g. how you calculated that mutational differences cannot explain the haplotype. What is the mutation rate you used? Also, you say that domestication history of the Chinese domestic geese and selective sweep events cannot explain the haplotype, so could explain this reasoning a bit more. As loss of polymorphism around the selected locus characterizes a sweep region, so why hard sweep was ruled out as a possible explanation? Leucistic geese (white plumage) are of course very rare in nature but such geese are observed in the wild. Thus, white geese could appear in the domestic grey goose population and there would be strong artificial selection for this attractive phenotype. In domestic animal population the effective population sizes can be very low, which could lead to a rapid fixation of the sweep region. This seems more discussion than results, please move this part to Discussion section. Finding of such haplotype block with very low polymorphism in white geese is very interesting.

Line 357-358: You cannot actually prove that the 14-bp insertion in EDNRB2 is an ancient haplotype as the ancestor of Chinese domestic geese, the swan goose, does not possess this insertion nor any of the wild relative species.

Line 362: Yes, it is possible that the haplotype has introgressed from an extinct species. However, as the swan goose does not possess the white feather haplotype, the introgression must have happened most probably after domestication which would place the introgression event in the last 6000 years and thus it is not very "ancient". However, I don't think your analyses are sufficient to try to pinpoint potential introgression events and looking into the origin of the white feather phenotype would require additional introgression analyses, which are out of the scope of this manuscript. Thus, I suggest that you are more cautious before making definitive statements or you need to provide more information on how you determined introgression as the only alternative.

Line 368: Overlapped by zHp and Fst methods?

Line 388: I'm uncertain to which comparative genomic analysis are you referring to. Please re-write

Line 396: Related to my previous comments, "conservation biology" is out of context and you should mention conservation of genetic resources of indigenous breeds or something similar.

Line 402: Long evolutionary process? The evolutionary history of domestic species is relatively short, so please re-phase this sentence

Line 402-406: Can you really say that expansion of Eph/ephrin bidirectional signaling pathway in XGG had lead to sensitive and highly alert characteristics of XGG geese as all geese are alert animals. I don't think it is possible to draw such a straight line in here.

Line 411-412: Change "As the offspring of a migratory bird" to "As the Chinese domestic goose has descended from a migratory bird" or similar.

Line 432-433: You say that: "then evolved the current population genetic structure", was there in reality a lack of population structure between these varieties?

Line 440-445: Related to my previous comment on line 292-294, you cannot directly compare heterozygosity values of microsatellites and SNPs with each other due to different properties of these markers. Instead, the heterozygosity in indigenous Chinese domestic geese seems to be on rather healthy level compared to the greylag goose derivative, the Landaise goose. You can look also the SNP study by Heikkinen et al. 2020 (Long-Term Reciprocal Gene Flow in Wild and Domestic Geese Reveals Complex Domestication History) to compare heterozygosity values with different European domestic goose breeds as this study uses SNP data. Please re-write this section of your discussion. You can justify the need for conservation of indigenous goose breeds with the need to preserve genetic variation, as indigenous breeds may harbor unique genetic variation for e.g. disease tolerance or tolerance for different environmental conditions.

Line 477-479: Could you please state what good results mean? Did you observe no increase in inbreeding related values?

Line 492: I missed the mutation rate in the results, but was actually here. Please state the unit of the mutation rate, was it base pair per year or per generation or what. As I have previously commented, I'm a bit concerned about this statement as you cannot pinpoint the species or a ghost lineage from which this insertion could have introgressed.

Line 497-500: This sentence was not clear, please re-phrase.

Line 520: Change "involved" to "involving"

Line 528-529: Change "Genomic" to "genomic". Please provide reference for this or describe the extraction protocol in more detail.

Line 709-715: By this analysis you are searching for signs of selection between white and grey geese, not particularly of the white plumage because you identify also other alleles selected between the white and grey geese. I think you should be more careful about the difference between selective sweep analysis and GWAS (genome-wide association analysis), which is the method to discover association between certain genomic regions and a specific trait, such as the plumage color. Please re-phrase that you searched for signs of selection between white and grey geese and then focused further analyses on one gene of interest, which has been linked to plumage or coat color in other species.

Fig.3 legend:

Line 1022: Change "anser" to "Anser". Also, I didn't note that there was Cygnus in the panel C. Please remove "cygnus" and place in the text describing panel D.

Table2:

Line 1044: Please add to the note explanation what the breed abbreviations in the table mean.

Reviewer 1:

This paper explores a consider number of goose resequencing data, comparative genomics and evolutionary analysis. However, these lots of analyses focus too much information which hinders the manuscript cannot answer one question in more detail. Analyses cannot fully match the aims of this study.

The first aim of this study is to study the phylogenetic position and evolutionary history of domestic goose. However, the background didn't give the sufficient reason for this aim. Previous studies explore the goose genome and comparative studies (Lu et al, 2013). Current study should explain why and how to improve the domestication events of goose. The results also focus the novel findings and genetic relationship of aisa goose and geese in other countries.

The 2,3 and 4 aims focused on the genetic structure of Jiangxi geese, and conservation strategy. But the results described too much un-related information about the conservation strategy, i.e., sex chromosomes, genome assembly. The positive selection events in the XGG didn't answer the local adaptability of the XGG and environmental relationship.

In terms of the conservation issue, authors stated "we then developed a selection principle in which recurrent cross-breeding of ganders among families and female geese were equally reserved for breeding offspring..." (line 315-320). These conclusions didn't provide the novel information, and also didn't do simulation of conversion, population genetic analysis for this so-called plan.

Lots of sequenced data are useful for bird biology and conservation study. However, too much aims of this study have been stated, and some conclusions are not accurate.

As stated above, this study did lots of work, but should focus on genome assembly improvement, comparative genomics and genomic structure of local geese. Or shorten these parts of analyses, then focus on the conservation strategy adjustment based on the molecular evidences.

Response: Thank you for your constructive comments on the improvement of this study. According to your advice, we have significantly revised the manuscript, especially the description of background (lines 38–96). The content of the conservation strategy has been removed, which will be reflected in our other study. The revised manuscript focuses on genome assembly improvement, comparative genomics and population genetics of Chinese indigenous geese. We hope that the revised manuscript addresses your concerns and that it is clarity and readability.

Reviewer 2:

This manuscript provides an in-depth work characterizing the geese species and breeds from a genomics perspective either in terms of comparative analyses as either in population genomics. These approaches reveal to be a novel and very comprehensive genomics characterization which unravels many important aspects in geese. For instance, the first goose genome that is not a hybrid; the sex chromosomes identified for the first time; among others.

There is quite a lot of work done and I even thought it could be partitioned into several articles that could also provide the opportunity to improve and deepen the specific results. In

fact, this amount of work makes it difficult to understand the flow or how the analyses are serving the mentioned objectives or motivation (and is also harder for the authors to organize so many things in a single manuscript).

Generally, I found that the text is well-written and it is visible that the authors have put a great effort to make it clear. However, I present some minor suggestions below to help improve clarity, line-by-line, that should be checked (whenever possible) throughout the main text, figures and supplementary material.

The presented methods and results are generally well conceived and presented, however, they need textual improvements for clarity and to ensure the reader is not lost by the lack of information or other aspects.

Response: We are grateful for your positive comments on this study, which greatly increases our confidence. We also greatly appreciate your constructive comments contribute to the improvement of our manuscript. We have done our best to improve the manuscript and have made many major revisions according to your comments. Too much content indeed makes difficult to understand for authors, we therefore partitioned this study into two articles. The content of the conservation strategy has been removed, which will be reflected in other study. The revised manuscript focuses on genome assembly improvement, comparative genomics and population genetics of Chinese indigenous geese. We hope that the revised manuscript addresses all questions and that it is more clarity and readability.

1. I feel that the manuscript title doesn't entirely make justice to the kind of work developed throughout the manuscript. Here, if possible, I would suggest to change to "Historical relatedness, conservation status, comparative and population genomics in the Chinese indigenous geese" (note "geese" here) or any alternative form which shows the most important genomics work that has been accomplished here, where (as a main part) and as stated in Discussion and other sections is "the first chromosome-level" of this breed and among all sequenced geese to date.

Response: Thank you for this valuable suggestion. Taking into account your comments and the content of revised manuscript, we have modified the manuscript title to "Chromosome-level genome and population genomics reveal evolutionary characteristics and conservation status of Chinese indigenous geese".

2. The Abstract section could also be improved to better describe the manuscript contents, maybe if the author's rewrite it following the organization into Background, Results and Conclusion subsections it may result in a better abstract (without necessarily containing these titles).

Response: Thank you for your advice, we have reworded the abstract section (lines 20–35).

3. In the Background section, I feel that a small paragraph by the end, would greatly help to quickly understand the manuscript layout, specifically the Methods and Results sections. This paragraph would be showing what work is being presented, or how the objectives are being

addressed by the several analyses, for instance, “In section X, we present these analyses (1,2,3...) to investigate this idea and/or to meet mentioned objective 1; second, we perform these analyses (1,2,3...) to investigate idea 2 and/or to meet mentioned objective 2; etc.”

Response: Thank you for this valuable suggestion. We have changed the ending paragraph of the Introduction section to “To provide a chromosome-level genome for Chinese indigenous geese, we used a hybrid de novo approach included PacBio, Illumina, 10× genomics, BioNano, and Hi-C technologies, and we combined with comparative genome analysis to explore the biological characteristics of XGG during the evolution of Anatidae. Large-scale resequencing of 994 geese was carried out for population genetic analysis to reveal the genetic diversity, genetic differentiation and resource conservation status. Additionally, we used selective sweep and allele frequency difference to detect the causal mutations and origin related to the plumage color of domestic geese, and the selection signatures of XGG population were also explored. Our study not only provides invaluable data resources for global geese research but also contributes to germplasm resource exploration, causal mutation of white plumage in Chinese domestic geese breeds, and geese breeding.” (lines 84–96).

4. As the Methods section is placed at the end of the manuscript it makes difficult to assess the Results (reading directly from Background) taking into account the different breeds, populations, species involved, and numbers used (e.g., 994, 990, 845, 257, 149, etc), so I believe that improvements to fill these gaps could help in terms of clarity and readability. This section should focus on describing the analyses, any problems faced and solutions. It should be clear how analyses are performed (software used, etc) and how data (types), numbers, etc are attained. Additionally, sometimes I felt that Methods or Results subsections could benefit some re-organization, as for instance, the positive selection analyses are not following the phylogeny analyses section, which breaks the necessary flow.

Response: Thank you for your suggestion. We have described the species involved and the number used in detail in the Results section and have revised the corresponding Methods section. We also re-organized and rephrased part of Methods and Results section according your comments below. For the positive selection subsection, we have re-analyzed and rephrased in light of your 100th comment (Methods: lines 574–584, Results: lines 204–218) are detailed in the revised manuscript.

5. The terms “assembly”, “chromosome-level” and “scaffolding” cause some confusion (See also comment in line 547 below). Each term pertains to a different step in establishing the genome contiguity. “Assembly” and similar, should be used to only refer to the process of reaching the contigs sequences from the sequencing reads. “Scaffolding” and similar, should be used for the process of reaching scaffolds sequences from the contigs sequences. And “chromosome-level” should be used when the scaffolds give rise to “chromosome-level” sequences like those in the Human genome. By reading the text (lines 99-156, and others) it becomes difficult to understand at which point is the reconstruction process of the genome or what is the term referring to. I understand that “assembly” is generally used to refer to the process of piecing together (and not necessarily the stage of reconstruction), but when there are other stages of reconstruction, it becomes confusing. If the authors write “assembly”

in instead of a synonym, it is perceived as only referring to this delimited step and any related analyses, but should not refer to others; and so forth. For instance, Lines 118-119, should the “assembled genome of XGG showed the largest scaffold” be modified into “scaffolded genome of XGG showed the largest scaffold”; Line 121: “our newly assembled genome” modified into “our newly scaffolded genome”? (For reference: <https://doi.org/10.1371/journal.pcbi.1006994>).

Response: Thank you for your correction. We have carefully checked the full text and made changes in the revised manuscript. Thank you for the detailed explanation of the step in establishing the genome. We apologies for the confuse description about this, we have made corresponding revisions for full text. we hope it is accurate now.

6. Line 21: In Discussion it is mentioned that this chromosome-level genome is constructed for the first time, why not here?

Response: Thank you. We have added “the first chromosome-level genome” in line 21.

7. Line 42: “charactics” is correct? Could “features” be a better word?

Response: Thank you. We have changed it to “features” in line 44.

8. Lines 48, 94: replace “goose” with “geese”?

Response: Thank you. We have changed “goose” to “geese” in revised manuscript.

9. Lines 50-55: Following readability and sections organization, I would suggest that the various geese breeds and anyother species from the study should be similarly and briefly described here (framing them also in terms of their original species/populations and domestic/wild; perhaps with the help of an additional main table) in addition to the current ones and any value for this study, like why are they important? It could be possible to move some information from Methods-Experimental animals section to this paragraph. Furthermore, they are also mentioned in Abstract, hence why not here?

Response: Thank you for your valuable advice. We have reorganized the section to make it more readable, added a brief description of studied geese breeds (lines 54–71), and added the main figure (Fig. 1) for clarity.

10. Line 75: five genes are here described and associated to plumage coloration in geese, however, there is a study, that mentions other genes being studied regarding plumage coloration (<https://doi.org/10.1186/s12864-015-1924-3>). It appears from the whole article, that only EDNRB2 has been studied and found to be partly causal for coloration (lines 357-358). From these, TYR is common, and others are not mentioned. Has any in this set of 7 genes been found and could these genes (differences) also significantly influence plumage coloration?

Response: Thank you for your advice. We know that many genes affect feather color in one or different species, so we carefully checked our results and the literature that you mentioned above. *ASIP*, *OCA2*, *TYR*, *TYRP1*, *MC1R*, *MITF*, and *KIT*, which mentioned in the literature¹ and in the Introduction section^{2,3} of this study, were not significantly selected in a genome-wide contxt through zFst and zHp analyses. The haplotypes of these genes showed similarity

between white and gray geese (Fig. R). According to these results, we considered that *EDNRB2* may significantly affect the white plumage of Chinese domestic geese.

Fig. R. **Haplotype analysis of seven pigment-related genes.** Pink and gray bars represent the Chinese white geese and Chinese gray geese, respectively. The black box represents the haplotypes of the white geese population, and the beige and orange colors represent the high and low frequency alleles in the white geese, respectively.

11. Line 79: is not clear whether this refers to the same previously mentioned breeds or if these references refer to different ones.

Response: Thank you for pointing out. The two domestic geese genomes here refer to Sichuan white goose and Zhedong white goose, respectively. To avoid confusion, we have rephrased these sentences in lines 47–48.

12. Line 89: what kind of “relationship”?

Response: Thank you. This refers to the genetic relationship between Chinese domestic geese breeds, we have rephrased this sentence to make it clearer (lines 88–90).

13. Line 91: “origination” should read “origin”? Or another better word.

Response: Thank you. We have changed “origination” to “origin” in line 91.

14. Line 108: the term “reduplication” is not clear, please explain. What is its importance

for the scaffolding process?

Response: Thank you for your reminder. “reduplication” refers to the scaffolds after the high-quality Illumina reads polish in the previous step. To avoid repeated descriptions, we have rephrased this sentence in lines 108–112.

15. Line 112: *what is “mount rate”? Do the authors mean coverage breadth (<https://doi.org/10.1038/nrg3642>, <https://doi.org/10.1371/journal.pcbi.1006994>)? In fact, coverage and read depth can be used to mean the same, but it would make more sense to distinguish both by using different and more appropriate terms with sequencing depth applied to refer to (e.g., ~99,10x) cases and coverage applied to the percentage that refers to the amount of sequencing that spans the entire genome.*

Response: Thank you very much. “mount rate” refers to the ratio of anchored 39 pseudo-chromosomes to the 2,242 scaffolds. We have rephrased this sentence by “Totally, Hi-C linking information supported 1.13 Gb (97.65%) of scaffold sequences being anchored, ordered, and oriented to 39 pseudo-chromosomes” (lines 110–111) and hope that it is clarity now.

16. Lines 117-120: *please provide references for these studies, regarding mentioned “genomes of other geese”, “studies in birds” and the “16 avian draft genomes”.*

Response: Thank you for your reminder. “16 avian draft genomes” including genomes of “other geese” and “studies in birds”, which are downloaded from the NCBI database. We have added references (except unpublished) and genome assembly versions for these genomes in Supplementary Table S5.

17. Line 119: *kakapo bird, is not clear if is a goose or any other species. Should the scientific/latin name be provided? References?*

Response: Thank you for your suggestion. We have changed “kakapo” to “kakapo (*Strigops habroptila*)”, and added a relevant reference in the revised manuscript (line 118).

18. Line 121: *“of the typical Chinese local goose” refers to the previously mentioned “female XGG”? Please clarify.*

Response: Thank you for pointing out. We have changed “of the typical Chinese local goose” to “of XGG” in line 120.

19. Lines 123-124: *references needed.*

Response: Thank you for your reminder. We have cited relevant references in the revised manuscript (lines 122–123).

20. Line 125: *replace “genome assembly” with “scaffolded genome”?*

Response: Thank you for your suggestion. We have replaced.

21. Lines 126, 128: *“the XGG genome draft” the word ‘draft’ should not be used since the genome is in scaffold level? The terms “draft genome” are usually employed to mean low accuracy and/or initial reconstruction. See also other occurrences.*

Response: Thank you very much for pointing out the question. We have removed “draft” in the revised manuscript (lines 124–150).

22. Line 129: *reference missing for CEGMA [?]*.

Response: Thank you. We have added a reference (line 128).

23. Line 130: *“covered” means “identified” and/or “annotated”?*

Response: Thank you. The “covered” means “identified”, we have replaced “covered” with “identified” (line 129).

24. Line 103, 134 and others: *Supplementary material indication sometimes is given with “Supplementary” word and in others is not.*

Response: Thank you for your reminder. We have carefully checked the full text and indicated the Supplementary material with the word “Supplementary” in the revised manuscript.

25. Line 132: *First occurrence of “Tian fu goose”. It seems the only one in text and in figures that has no abbreviation, suggesting “TFG”. Please modify accordingly. Additionally, I suggest the use of the (existing) abbreviations should be much more common in text, figures, tables, etc to quickly identify which breed or species is being referred (e.g., Line 151: “TFG” could be used instead or in addition to “hybrid goose”).*

Response: Thank you for your advice. We have added the abbreviation “TFG” for Tian fu goose in line 46 and applied it in subsequent text.

26. Line 138: *missing reference for the mentioned “results of Tian fu goose”.*

Response: Thank you for your reminder. We have added a reference for the mentioned “results of Tian fu goose” in line 135.

27. Line 140-142 *needs rephrasing, it starts by giving a value associated to exons, and next values are suddenly given in tandem. It should be easier to read in “value item,” form.*

Response: Thank you for your suggestion. We have rephrased this sentence in the lines 135–138, presenting the values of average transcript length, etc. in Supplementary Table S8 in a tabular form, hopefully now clearer.

28. Lines 143-144: *“draft genome” and “chromosome-level goose genome”, which is which? Is “draft” appropriate term?*

Response: Thank you for your reminder. We do also recognize that the term “draft” is inappropriate in this study. According to your suggestion, we have modified the sentence to “The number of genes in XGG genome was close to that of TFG” in the revised manuscript (lines 138–139).

29. Line 149: *“folded” is the correct term? What it means?*

Response: Thank you. Repeat sequences are widely distributed in eukaryotic genomes, and these repeat sequences are either concentrated in clusters or scattered among genes. Most

notably, the repeat sequences are so similar that they collapse into one gene/region, displaying much higher coverage than the other region. It would be appeared some issues at the annotate process (i.e. annotated as a single gene while in reality multiple, or the genes might be hidden from annotation because the software registers them as repeats). Additionally, long tandem repeats (LTRs) range in length 1167–2726 bp. When the read length of the sequencing method is shorter than the LTR, repeat number can be massively misjudged, which can affect the identification of the results (doi: 10.1093/nar/gkz841). Hence, we use the word “folded” here.

30. Line 161: “three methods” which are? Or should it be “one method consisting of 3 steps?”
Replace “locate” with “identify”?

Response: Thank you for your reminder. We have changed “three methods” to “three steps”, and replaced “locate” with “identify” in lines 153–155.

31. Line 162: “of the typical Chinese indigenous goose” could be removed? Could the whole sentence be reduced to “To further explore such genomic characteristics in XGG, we integrated three steps to accurately identify the sex chromosomes, which has not been accomplished in hybrid goose genome [15].” Feel free to improve.

Response: Thank you for your constructive suggestion. We have changed this (lines 153–155).

32. Line 164: “panel” word could be replaced by “reference” or “control”?

Response: Thank you for your advice. We have replaced “panel” with “reference” in line 157.

33. Line 166: delete first “and” word. Are these the previously mentioned “three methods”?

Response: Thank you for your careful review. We have deleted the first “and” word in line 157. “sequence splitting, homology alignment, and classification” refers to the previously mentioned “three methods”, and the “three methods” has been revised to “three steps”.

34. Line 172: a word is missing: “closer”? Also from the Peking duck?

Response: Thank you for your reminder. We have added “closer” word in line 166 and modified the sentence to “while Hic_4 presented a closer synteny to the W chromosome of Peking duck” in lines 164–165.

35. What is the correct spelling of Peking Duck with “g” or without? It appears to be misspelled in several occurrences, whereas in Fig 1d appears with “g” but not in Fig 1a and in the main text (Line 171, etc).

Response: Thank you for your careful review. According to the relevant literature^{4, 5}, we consider “Pekin” may be the correct spelling. We have carefully checked the full text and replaced “Peking” with “Pekin” in the revised manuscript.

36. Lines 172-176: the involvement of 103 females and 59 males is not previously explained, how this happened and why? Which breeds?

Response: Thank you for your comments. The involvement of 103 females and 59 males included 50 LXW, 51 FCG, 50 GFW, and 11 LHW, and these whole genome resequenced individuals were from those used in the following population genetic analysis. The specific information of the 103 females and 59 males are shown in Supplementary Table S15. Since these geese accurately recorded sex information during sampling, we used it here to assess the accuracy of sex chromosome identification.

37. Line 186: replace “applied” with “employed”?

Response: Thanks for your suggestion. We have changed “applied” to “employed” (line 176).

38. Lines 196, 200, 202: which “goose and chicken”? Plural?

Response: Thank you for your reminder. "Goose and chicken" is a general term here. According to your comments, we changed them to plural form (lines 184, 188, and 189).

39. Line 204: “four goose genomes” which are, please specify? Would “geese” be more correct?

Response: Thank you for your careful review. We have changed “four goose genomes” to “four geese genomes (XGG, SCW, ZDW, and TFG)” in the revised manuscript (line 192).

40. Line 205: provide figure identification.

Response: Thank you for your reminder. We have added figure identification “Supplementary Fig. S7” in line 194.

41. Line 206: “and” should read “of which”? Is it 11,648+15?

Response: Thank you. We have changed “and” to “of which” in line 193.

42. Line 206: “containing” replaced by “including”?

Response: Thank you. We have changed “containing” to “including” in line 194.

43. Line 207: “gene” word is missing.

Response: Thank you. We have changed “families” to “gene families” in line 195.

44. Line 210: the figure 1d shows 131 expanded and 38 contracted gene families, is there also any idea of which are the 38 gene families contracted in XGG (plus number of genes) and possible effects?

Response: Thank you for your reminder. We were also interested in the possible effects of the 38 gene families contracted in XGG. Enrichment analyses were also performed on these contracted gene families and referred to relevant literatures to explore the possible functions of these genes. Unfortunately, we did not enrich any entries, so we were not mentioned in the manuscript.

45. Line 222: The id GO:0051186 is shown as obsolete, is there any alternative GO?

Response: Thank you for your reminder. We apologize for the carelessness about the obsolete id GO:0051186. We have performed GO enrichment analysis on the positively

selected genes again according to your 100th comment. Corrected results are shown in Supplementary Table S20, and rephrased this in the revised manuscript (lines 204–218).

46. *Line 226: missing citation for GO.*

Response: Thank you for your reminder. We have added the reference for GO and KEGG in line 209.

47. *Line 226-234: Please clarify this sentence. Are you referring to aforementioned GO terms or genes? Is FAS another GO id? Is this related to the positive selection detected?*

Response: Thank you for your reminder. We have rephrased this in lines 204–218.

48. *Line 236: What proportion or number of individuals pertains to each population/breed?*

Response: Thank you for your advice. We have added the number of each population/breed on the lines 220–223.

49. *Line 239: the 772 are XGG and the 222 are? “1X” and “10X” should specify their meaning.*

Response: Thank you for your advice. 222 geese including 51 FCG, 50 GFW, 11 LHW, 50 LXW, 5 ACy, 5 AAn, and 50 LDG, which have been revised in the manuscript. The specific meanings of “1X” and “10X” have been added. (lines 224–229)

50. *Line 246: “Eventually” should be replaced or removed. 845 geese are all XGG?*

Response: Thank you for your advice. We have removed “Eventually”. We have added the detail of 845 geese “(including 636 XGG, 50 FCG, 46 GFW, 9 LHW, 45 LXW, 49 LDG, 5 ACy, and 5 AAn)” in the revised manuscript (lines 236–237).

51. *Line 248: the number of SNPs can be removed, just say “Among these SNPs”*

Response: Thank you. We have changed “Among the 11,029,910 SNPs” to “Among these SNPs” in line 238.

52. *Line 254: the 845 individuals are the same as the 845 from previous section?*

“neighbor-joining (NJ) tree” is abbreviated here for the first time, but only used once (line 703) throughout the text. It is recommended the use distinguishing abbreviations for different NJ trees / methods.

Response: Thank you for your reminder. The 845 individuals are the same as the 845 from previous section. We have replaced “845 individuals” with “845 geese” in the revised manuscript to avoid confusion (line 242). For the abbreviation “NJ” of “neighbor-joining”, we used this abbreviation (NJ) in the following text (lines 255 and 611) and added the corresponding figure identification.

53. *Line 264-265: It becomes unclear if the new dataset of 257 was important for the next paragraph or if this still results from previous analyses. From the previous two sentences it is not clear how the 257 dataset was attained and what breeds are included. I would suggest*

to start the following sentence (l.265) with “Here,” instead of “Similarly,”.

Response: Thank you for your constructive suggestion. The new dataset of 257 geese was used for the population analyses shown in Fig. 3, we reworded this section in the lines 248–263, hopefully it is clear now. “Similarly,” has also been replaced by “Here,” in the revised manuscript (line 252).

54. Line 281: why not 845 or even 257 individuals instead of 990? Were these four individuals also removed from previous analyses?

Response: Thank you. To truly reflect the genetic diversity of the populations, we use as much individuals as possible to estimate genetic diversity parameters. A total of 994 samples were collected in this study, and four outliers of XGG were found in previous quality control, possibly due to recording errors during the sampling process. Therefore, we removed these four samples here, which were removed in previous and follow-up analyses.

55. Lines 283 and 294: it seems both LDG and LHW had the “lowest genetic diversity”? Please clarify.

Response: Thank you for your reminder. We have rephrased this section in lines 265–282.

56. Lines 306-307: which one is this NJ tree, the Fig. S7 or Fig. 2B?

Response: Thank you. This part belongs to the conservation strategy study, according to your previous comment of “There is quite a lot of work done and I even thought it could be partitioned into several articles” and comments from other reviewers, we have removed the conservation strategy study, which will be reflected in our other studies.

57. Line 333: Table S24 shows gene EDNRB, but is this different from EDBRB2?

Response: Thank you for your careful review. This is due to our carelessness, actually it should be EDNRB2 here, which we have corrected in Table S22.

58. Line 345: “presented” or “present”?

Response: Thank you. We have rephrased this part in lines 304–307.

59. Lines 378-380: I am not sure if “usually” is the right word. Diseases can also be onset from several non-immune related genetic defects (affecting DNA, RNAs, proteins, etc). Cysts are often benign or noncancerous, hence to write that they may be related to immune genes, requires evidence. Is there any study or analysis that can confirm or point to the association of these cysts with immune-related genes?

Response: Thank you for your valuable suggestion. Recent article titled “Transcriptome analysis of Echinococcus granulosus sensu stricto protoscoleces reveals differences in immune modulation gene expression between cysts found in cattle and sheep” suggested that there is an association between cysts and immunomodulatory genes in animals, so we speculated that the cysts in the goose foot might be immunologically related. We have added this reference in the revised manuscript (line 333). To avoid confusion, we removed “usually” and rephrased this sentence to “Immunity is the organism’s own defense mechanism, and the onset of some diseases are immune related” (lines 332–333).

60. Lines 390-393: To identify the function, have the authors performed homology searches of this gene in databases, for instance in NCBI? Do the results confirm the speculation?

Response: Thanks for your valuable comments. We performed homology searches of Hic_asm_9.361 gene in the NCBI database and found that the gene is as high as 88% homologous to the chicken *CLCA1* gene, so we made corresponding changes in the main text (lines 342–347).

61. Line 423: the LXW first occurrence is in line 267 (no previous or current abbreviation), but the indigenous geese from Jiangxi province were described in lines 50-55. Should the LXW also be included here?

Response: Thank you for your reminder. We have added LXW and its full name in line 60.

62. Line 425: “chronicles” could mean “literature”?

Response: Thank you. We have changed “chronicles” to “literature” in line 383.

63. Line 425-430: following the previous point (line 423), should this (and other) sentences be moved (or reused) to Background section near the mentioned paragraph? See also comments in lines 50-55.

Response: Thank you for your valuable advice. We have added the relevant background in lines 54–71.

64. Lines 497-500: this sentence should be rephrased. Did you mean: added by accident?

Response: Thank you for your reminder. We have rephrased this sentence in lines 429–434.

65. Lines 515-517: “and the data fill gaps in our knowledge and facilitate further “could be replaced by “and the results further facilitate”?

Response: Thank for your advice. We have replaced “and the data fill gaps in our knowledge and facilitate further” with “and the results further facilitate” in line 446.

66. Lines 532-533: why this separation of 994 into 772 and 222? What took to these numbers? (See also line 651)

Response: Thank you. It is undeniable that the higher the sequencing depth, the more accurate the result of SNP calling. However, considering the sequencing cost, research purpose and the currently mature genotype imputation algorithm, we performed low-depth sequencing ($\sim 1\times$) on all 772 XGG and then imputed them to genome-wide level, the remaining geese (222) were sequenced at $\sim 10\times$.

67. Line 535: this adult female is the same individual as the one mentioned in line 530? Should this description be moved to the above section (or Background)? See also comments in lines 50-55.

Response: Thank you for your reminder. The Methods section has been reorganized, the adult female is the same individual as the one mentioned above, we have moved it to the lines 463–464.

68. Lines 538-545: should these values be repeated (here and in Results)?

Response: Thank you for your reminder. We have rephrased these sentences in lines 469–480.

69. Line 539-541: how were the Super-scaffolds reconstructed? Which software (no citation)? Wtdbg2 version missing.

Response: Thank you for your comments. We have added the Super-scaffolds reconstructed process, corresponding software and its version in the lines 471–474. The version of wtdbg2 is added in line 469.

70. Line 541: the “preliminary assembly” refers to the PacBio assembly or to the resulting scaffolds, upon applying the linked-reads from 10x platform?

Response: Thank you for your comments. The “Preliminary assembly” refers to scaffolds generated from Pacbio-assembled contigs combined with link-reads from the 10× platform. To avoid confusion, we have rephrased this part in lines 471–480.

71. Line 542: the “assembled version” refers to the scaffolds upon being filled by Bionano data?

Response: Thank you for your comments. The “assembled version” refers to the scaffolds upon being filled by BioNano data. To avoid confusion, we have rephrased this part in lines 474–476.

72. Lines 541-542: how were the gaps filled? Which software (no citation)?

Response: Thank you for pointing out. We have added the gap filling software in line 475.

73. Line 544: Pilon version missing.

Response: Thank you for your reminder. We have added a version for Pilon software in line 477.

74. Line 545: what was combined with Hi-C data?

Response: Thank you for pointing out. We have modified this sentence in line 478.

75. Line 546: LACHESIS version missing.

Response: Thank you for your reminder. We have added a version for LACHESIS software in line 479.

76. Line 547: “chromosome-level scaffolds” are these sequences chromosomes (like those in the Human genome) or still scaffolds? In case these are not exactly chromosomes, I would suggest to start refer to them as either scaffolds (or super-scaffolds) or eventually “near chromosome-level sequences”. In supplementary material the term “pseudo-chromosome” is used, is this with same meaning?

Response: Thank you for your constructive suggestions. Here, we adopted Hi-C technology to anchor, order, and orient 1.13 Gb of sequences into 39 chromosomes, so we have replaced

“chromosome-level scaffolds” with “chromosome-level sequences” in line 480. However, since further validation was not performed using PacBio HiFi in combination with extensive FISH and cytogenetic experiments, a "pseudo-chromosome" was used in the supplementary material.

77. Line 548: “assembled” could be replaced by “reconstructed” or “scaffolded”?

Response: Thank you. We have replaced “assembled” with “reconstructed” in line 480.

78. Line 549: Was CEGMA used for all this step or in addition to...? Please improve. Software version missing.

Response: Thank you for your reminder. CEGMA is used for the assessment of genome assembly integrity, we have rephrased this sentence in the lines 479–483, the version of CEGMA is also added here.

79. Line 559: Repbase needs a citation.

Response: Thank you for your reminder. We have added a reference for Repbase in line 492.

80. Line 560: RepeatMasker version missing.

Response: Thank you. We added a version for RepeatMasker software in line 493.

81. Line 561 and others: Careful should be taken when writing “gene prediction” or “prediction” as opposed to “gene identification”. Gene predictions focus on detecting novel gene structures or models and any regulatory regions, which tend to have not previously been detected (E.g., GenScan (<https://doi.org/10.1006/jmbi.1997.0951>), Genefinder, Genewise (<https://doi.org/10.1101/gr.1865504>). Whereas gene identification would be a better term for simply identifying genes that are à priori known in other species, using strategies like those employing BLAST and any of the above.

Response: Thank you for your valuable comments. We have changed “prediction” to “identification” in lines 494–496, and also carefully checked the full text in the revised manuscript.

82. Line 565: which was the software used, BLAST or BLAT? If the former, then the reference needs to be adjusted accordingly. Could <https://doi.org/10.1186/1471-2105-10-421> this be appropriate? Software version missing.

Response: Thank you for your reminder. The software used here is BLAST, we have corrected references and added a version for BLAST in line 499.

82. Line 566: “genewise” would be better written as “GeneWise”. Version missing.

Response: Thank you for your advice. We have changed “genewise” to “GeneWise” and added the version in line 499.

83. Lines 566-568: GeneWise is used to predict gene structure and the following three are used for “de novo prediction.”. Are all four employed for the same purpose? GlimmerHMM version missing.

Response: Thank you for your careful review. All software were used for gene structure identification and prediction, with GeneWise for homology-based identification and others for *de novo* prediction, so we have rephrased this sentence in lines 496–501. We have added a version for GlimmerHMM software in line 500.

84. Lines 568-570: *this sentence needs rephrasing. “genes were annotated by the prediction results”, how and why? Functional annotation? Why “combined with RNA-seq comparison data”? Did this help to identify genes or extra data was used in the process? Which “comparison data” was used? Software version missing.*

Response: Thank you for your reminder. We have rephrased this sentence in lines 501–508 and added corresponding software version in the revised manuscript.

85. Line 571: “NR” should be better written as “NCBI nr”? Citations are missing in all cases.

Response: Thank you. We have changed “NR” to “NCBI nr”, and have added citation (line 509).

86. Line 576: *replace “for reference, including” with “to be used as reference sequences. These included”? From which databases were the sequences downloaded? Provide citations and any sequence IDs or accessions in Table S13.*

Response: Thank you for your advice. We have replaced “for reference, including” with “to be used as reference sequences. These included” in line 514. These sequences were downloaded from the NCBI database, and we have provided references and corresponding version in Supplementary Table S13.

87. Line 579: *“the” is missing. SAMtools version missing.*

Response: Thank you. We have added a version for SAMtools software in line 517.

88. Lines 583-584: *“the scaffolds” repeats.*

Response: Thank you for your reminder. We have removed the second “the scaffolds” in the revised manuscript.

89. Line 586-587: *why TBtools is not described in the detailed Method S3? Version is missing. Suggest moving “for accuracy assessment,” to the start of the sentence.*

Response: Thank you for your suggestion. We have added the detailed TBtools method in Supplementary Method S3, the version of this software is also added in the revised manuscript (line 525). According to your suggestion, we have moved “for accuracy assessment,” to the start of the sentence (line 524).

90. Line 588-589: *how were the depths calculated? Which software (no citation)?*

Response: Thank you for your comment. The sequencing depth was calculated by SAMtools (option: --depth). We have rephrased this sentence in lines 526–530 and added a citation for SAMtools.

91. Line 596: *which protein sequences were used? All? Did this included the identified sex*

chromosomes?

Response: Thank you. We used all protein sequences here, including those from the identified sex chromosomes.

92. *Lines 600-602: Did the clusters help to determine the gene families? How?*

Response: Thank you for your comment. Gene families were identified using OrthoFinder v2.4.0. Clusters here refer to gene families, to avoid confusion we changed “clusters” to “gene families” in lines 539–540.

93. *Line 604: why 10 species?*

Response: Thank you for your comment. Here, we used protein sequences to obtain single-copy orthologous genes for comparative genomic analysis and reconstruct a phylogenetic tree. Since many species of *Anatidae* do not provide protein sequences or annotated files in public databases, we were unable to obtain additional protein sequences at the time. We downloaded as many representative bird species as possible from *Anatidae*. We have rephrased this sentence for more clarity (lines 543–544).

94. *Line 607: what does “self-blast” mean? Was BLAST used? Was BLAST also used in above subsection?*

Response: Thank you for your comment. We utilized OrthoFinder v2.4.0 to identify orthologous gene families, which is integrated software that automatically calls protein sequence alignment software (eg., BLAST and diamond). The older version of OrthoFinder was to call BLAST, while OrthoFinder v2.4.0 used diamond for protein sequence alignment. To avoid confusion, we corrected this sentence in the revised manuscript (lines 545–547).

95. *Line 618: PAML version missing.*

Response: Thank you. We have added a version for PAML software in line 557.

96. *Line 629: How was the number of 16,055 orthologous gene families reached?*

Response: Thank you for your reminder. The 16,055 orthologous gene families were identified by OrthoFinder v2.4.0 from 12 species used in the phylogenetic tree. We have modified this sentence in lines 567–569.

97. *Lines 635-638: this seems to be the same methodology used in phylogenetics analyses, could this be resumed to a simplified reference to the section and the resulting phylogeny and alignments? Were the concatenated MSA and tree used in studying positive selection?*

Response: Thank you for your reminder. Here, we performed positive selection analyses for each gene based on the species tree generated by concatenated MSA. Also, according to your 100th comment, we generated gene trees for each single-copy gene for positive selection analysis. The modified method is detailed in lines 574–584.

98. *Line 639: “EMBOS” is better written as “EMBOSS”.*

Response: Thank you. We modified the method for positive selection analysis, EMBOSS software was not used in the revised method.

99. Line 640: “paml” is better written as “PAML”.

Response: Thank you for your reminder. We have modified “paml” to “PAML” in line 577.

100. Lines 638-646: *The positive selection analyses should be performed for each DNA gene and to this end, each MSA and corresponding phylogenetic tree, should be estimated based on the same orthologous sequence dataset. Despite the important back-translation process, can the authors ensure the whole procedure is correct? Due to the degeneracy of genetic code, is it ensured that the original codons are being used for both cases? Or, instead could you download and use the (protein-corresponding) original DNA sequences? Why wasn't a DNA-based phylogenetic tree estimated for each gene? Additionally, the study (<https://doi.org/10.1093/molbev/msq303>) mentions that these types of analyses are suitable for detecting episodic positive selection which affects only a subset of codons, why the authors opted for branch-site models and not other, for instance, the site-models? Would different models results alter the manuscript conclusions relative to this part? Here I would also suggest the authors to consider the read and/or use of LMAP and LMAP_S software published in BMC Bioinformatics.*

Response: Thank you for your constructive comments. We have re-performed the positive selection analysis for each single-copy orthologous gene. Despite the important back-translation process, we cannot avoid the problem of codon degeneracy. Therefore, we downloaded and used the (protein-corresponding) original DNA sequences to estimate a DNA-based phylogenetic tree of each gene by IQ-TREE v.2.1.1, and performed positive selection analyses based on the branch-site model of PAML 4.9j.

The ω ratio is a measure of natural selection acting on the protein. Simplistically, values of $\omega < 1$, $= 1$, and > 1 means negative purifying selection, neutral evolution, and positive selection. However, the ratio averaged over all sites and all lineages is almost never > 1 , since positive selection is unlikely to affect all sites over prolonged time. Thus interest has been focused on detecting positive selection that affects only some lineages or some sites. The branch-site models allow ω to vary both among sites in the protein and across branches on the tree and aim to detect positive selection affecting a few sites along particular lineages (called foreground branches; <http://abacus.gene.ucl.ac.uk/software/pamlDOC.pdf>). The purpose of this part is to explore the positive selection of the Xingguo gray goose lineage in the evolution process, so we choose the branch-site model here. The revised method showed in lines 574–584.

101. Line 648: *replace “implemented” with “processed” or other better word.*

Response: Thank you for your suggestion. We have replaced “implemented” with “processed” in line 586.

102. Line 653: *Sentieon version missing?*

Response: Thank you. We have added the version of Sentieon software in line 593.

103. Lines 651, 659, 660: *should indicate what the (1X, 10X) are.*

Response: Thank you for your suggestion. We have added the meaning of 1X and 10X in

lines 588–591.

104. Lines 651, 659: *I find it odd that the authors have divided into a dataset of 772 XGG only and 222 of diverse breeds. Moreover, they are sequenced at different depths, why? Could the highest number of detected SNPs be due to the highest breed diversity in 222 datasets?*

Response: Thank you for your comments. Considering the cost of whole-genome resequence and the genotype imputation algorithm of STITCH, we performed low-depth sequencing ($\sim 1\times$) on all 772 XGG and then imputed them to genome-wide level, the remaining geese (222) were sequenced at $\sim 10\times$.

105. Line 664: *SnEff version missing.*

Response: Thank you for your reminder. We have added the version for SnpEff software in line 603.

106. Line 667: *--genome is an option not a command, unless “plink --genome” is the complete command. Hereafter, the software options and commands start to appear in the main text. They should all be discriminated in full for all cases and analyses and not just for a few. Hence, I suggest to include (and move) all of them (present and omitted) in Methods supplementary materials files in the appropriate sections. This way also avoiding to increase the manuscript length. This should also include software with graphical interfaces, by indicating the all main functions used.*

Response: Thank you for your suggestion. We have changed “command” to “option” (line 603). At the same time, the full text and supplementary materials has been checked, and then the corresponding places have been revised. We also submitted all scripts used to analyse the data, from genome assembly through to downstream population-level analyses to zenodo website (<https://zenodo.org/>; doi:10.5281/zenodo.6613753; lines 683–684).

107. Line 675: *“phylip” should be written “PHYLIP”.*

Response: Thank you for your reminder. We have changed “phylip” to “PHYLIP” in line 612.

108. Line 676: *Figtree is missing citation (or as URL).*

Response: Thank you for your reminder. We have added the URL of Figtree in line 611.

109. Line 682: *ADMIXTURE has a version missing?*

Response: Thank you for your reminder. We have added the version of ADMIXTURE software in line 619.

110. Lines 693, 697: *no need to repeat software versions, here and elsewhere.*

Response: Thank you for your reminder. We have removed the software versions (lines 628 and 634). We carefully checked the full text and removed repeat software versions in the revised manuscript.

111. Line 701: *This sentence could be reduced to refer to previous section.*

Response: Thank you for your advice. This part is the methodology of conservation strategy study, according to your previous comments, we have removed the conservation strategy study, which will be reflected in our other studies.

112. Line 703: “*phylip*” should be written “*PHYLIP*” and version number at first occurrence, unless different versions were used.

Response: Thank you for your advice. We have removed this section.

113. Lines 713-714: why these numbers?

Response: Thank you. To reduce the false positive and false negative may caused by the large difference in the number of groups (white and gray geese), we randomly selected 50 XGG and then combined them with other Chinese geese (100 gray geese VS. 109 white geese) to perform selective sweeps analysis.

114. Line 718: “*language*” can be deleted. Alternatively, a citation should be used for *R*.

Response: Thank you for your advice. We have deleted “*language*” in line 647.

115. Line 736: “*pegas*” should have a version number?

Response: Thank you for your reminder. We have added the version of *pegas* in line 664.

116. Lines 738-747: are these related to section starting in line 709? Why are they separate?

Response: Thank you for your comments. These two parts are the content of the selection sweeps, the above section is “Selection signatures between Chinese gray and white geese”, which explores the feather color of Chinese domestic geese. And this section mainly explores the selection characteristics of Xingguo gray geese. According to the writing order of the results section, we separate the methods of these two parts. To avoid redundancy, we combined these two sections in the revised manuscript (lines 640–643).

117. Line 741-742: suggest moving “to characterize the germplasm characteristics of XGG”, to the beginning of sentence?

Response: Thank you for your advice. We have moved “to characterize the germplasm characteristics of XGG” to the beginning of sentence (line 640).

118. Overall figures are good, but captions should be checked accordingly to previous detailed comments. Figure 2a could benefit from arranging the geese figures next to the points marked on the map (thus removing the doubled abbreviations). As it is, does not look like a legend. Or, perhaps the legends from a) and e) could be unified into a single legend. Even though, Figure 2a would make more sense as an initial figure to show the breeds being described (i.e., since Background lines 50-55).

Response: Thank you for your constructive suggestion. We have removed the double abbreviations and splitted Figure 2a as the initial figure (Fig .1) showing the breeds described.

119. Overall tables are good, but legends and titles should be checked accordingly to previous detailed comments.

Response: Thank you for your advice. We have carefully checked the full text and modified the legends and titles based on previous detailed comments.

120. It is visible that many of the referenced software and other material is not referenced in main text, which also makes hard to fully understand the Methods section. Ideally, they all should at minimum be enumerated, or at least those that are not at the detail level.

Response: Thank you for your advice. We have carefully checked the full text and revised it as much as possible.

121. Line 42: hybridScaffold requires citation.

Response: Thank you for your reminder. We have added the reference of hybridScaffold in line 44.

122. Line 50: BWA requires citation.

Response: Thank you for your reminder. We have added the reference of BWA in line 52.

123. Lines 53-79: In Stage 5, it is confusing because the type of data from Hi-C technology is not mentioned and the stage 4 data supposedly in scaffold form is not mentioned. Thus taking to the idea that the data being used is from previous stages, is this correct?

Response: Thank you for your reminder. We have rephrased the content of stage 5 in lines 57-83.

124. Line 92: the study mentions the capture of material for transcriptome (main text, lines 530-531; 569) and here the “assembled transcriptome” is mentioned (and further in lines 127, 138), but so far no section has described these methods? Are they related?

Response: Thank you for your comments. Transcriptome material is used for transcriptome sequencing to assist genome annotation. This section of the method has been added in the revised manuscript (lines 458–459 and 501–506).

125. Line 108: TRF (Tandem Repeats Finder) should be unabbreviated and cited.

Response: Thank you for your advice. We have added a full name and citation for TRF in line 113.

126. Line 118: uclust needs a citation.

Response: Thank you for your reminder. We have added the citation of uclust in line 123.

127. Line 130: BLAST could have a different citation (please see above), or in addition to this.

Response: Thank you for your reminder. We carefully checked the full text and made a uniform citation to BLAST (line 136).

128. Line 143: EVM and PASA are the same citation, but should also be cited next to PASA.

Response: Thank you for your reminder. We have added the citation next to PASA in line 151.

129. Line 147: *BLASTp* should be cited. Should use <https://doi.org/10.1093/nar/25.17.3389>

Response: Thank you for your reminder. We have added the citation in line 155.

130. Lines 148-150: missing citations. See also main text line 571.

Response: Thank you for your reminder. We have added citations to these items in lines 156–158, also modified in the main text.

131. Line 155: missing citation for *GO*.

Response: Thank you for your reminder. We have added a citation for *GO* in line 164.

132. Line 161: “blast” is an abbreviation and should be written as *BLAST*. This should be enforced for all occurrences (even to file formats), i.e., correct spelling.

Response: Thank you for your constructive advice. We have carefully checked the full text and changed “blast” to “BLAST” in line 170.

133. Line 181: How was quality control achieved? Used any software?

Response: Thank you. Quality control refers to eliminate the aligned reads with Mapping Quality < 30 through SAMtools, described in line 191.

134. Line 203: which “above reference genome”? Please specify.

Response: Thank you for your advice. We have changed “above reference genome” to “XGG” in line 215.

135. Line 204: “sam” file format is an abbreviation.

Response: Thank you. We have added a full name in line 216.

136. Line 205: “bam” file format is an abbreviation.

Response: Thank you. We have changed “bam” to “BAM” in line 217.

137. Line 206: PCR unabbreviation required. Where did this come from?

Response: Thank you. We added the full name of PCR (lines 218–219), which was generated by the Illumina sequencing process.

138. Line 207, 209: “Haplotyper” and “GVCFTyper” are software or functions of Sentieon? Citations might be missing. “gvcf” is an abbreviation.

Response: Thank you for your reminder. “Haplotyper” and “GVCFTyper” are the function of Sentieon software. We changed “gvcf” to “GVCF” and added its full name in line 220.

139. Line 209: *bcftools* requires citation. Should be written as “*BCFtools*”.

Response: Thank you for your reminder. We have changed “bcftools” to “BCFtools” and added the citation in line 222.

140. Lines 211-213, 219-220: What is the meaning of “||”? Should be translated accordingly.

What is the meaning of these metrics? A list of abbreviations could help establish the meanings for each abbreviation in main text and related files.

Response: Thank you. We have changed “||” to “,” and added the meaning of these metrics in lines 225–229.

141. Lines 213, 217, 220: “maf” is Minor Allele Frequency (MAF)?

Response: Yes. We have changed “maf” to “MAF” and added the full name when it first appeared in lines 228–229.

142. Table S1: It seems peculiar that only two sequencing technologies have insert sizes specified. In Line 6, it is mentioned a 20 Kb insert, but is missing in table. Are others missing too? And read-length?

Response: Thank you for your comments. We have supplemented Supplementary Table S1 with corresponding insert library sizes and read lengths. Since other sequencing technologies use libraries of different lengths, they are not indicated in the table.

143. Table S3: What is the difference between both (bp) values?

Response: Thank you. We have replaced “Assembly number” with “Number of anchored bases”, which refers to the bases located on the chromosomes. “Total number of bases” includes bases located on the chromosomes and scaffolds.

144. Table S4: Table title, “Base” could be modified to “Nucleotide”.

Response: Thank you for your advice. We have modified “Base” to “Nucleotide”.

145. Table S8: It is confusing to have one column with both species and software. The column on the right also needs better title. Software should be cited.

Response: Thank you for your reminder. We have modified the title of the columns on the right and added citations to the softwares in Supplementary Table S8.

146. Table S9: Needs improved titles in columns.

Response: Thank you for your reminder. We have modified the titles of columns in Supplementary Table S9.

147. Table S10: Needs improved title in “Type” column, like “Software” or “Method”.

Response: Thank you for your advice. We have changed “Type” to “Method” in the title of Supplementary Table S10.

148. Table S12: Needs improved titles in columns.

Response: Thank you for your reminder. We have modified the titles of columns in Supplementary Table S12.

149. Table S14: Needs improved title.

Response: Thank you for your reminder. We have modified the title of Supplementary Table S14.

150. Table S15: as mentioned in the Note, this table is showing sequencing depth for each individual and their chromosomes. Title can be improved and Note could be deleted.

Response: Thank you for your advice. We revised the title of Supplementary Table S15 and removed the note.

151. Table S18, S19 and S20: “P value” should be written as *p*-value, here and elsewhere.

Response: Thank you for your reminder. We have changed “P value” to “*p*-value” (Supplementary Tables S17–20) and carefully checked the full text and corrected it in the revised manuscript.

152. Table S19: Gene Symbols should also be upper-case as done with Table S21.

Response: Thank you for your reminder. These Gene Symbols have been changed to upper-case in Supplementary Table S19.

153. Table S22: Needs improved titles in 3 columns.

Response: Thank you. We have changed.

154. Table S25: Allele1 and Allele2: is one considered the reference allele? Which one? NCHROBS title could be clearer.

Response: Thank you for your reminder. To more intuitively reflect the content of the table, we have changed “Allele1” to “Alternative allele”, “Allele2” to “Reference allele”, and “NCHROBS” to “Number of alleles” in the revised Supplementary Table S23.

155. Line 393: spelling.

Response: Thank you. We have changed.

156. Figure S3: This figure mentions Trinity software, but not found in any text. Was it used?

Response: Thank you for your careful review. Here, we do not use Trinity software and we have corrected Fig. S3.

#Reviewer 3:

Ouyang et al. have constructed a chromosome-level reference genome of Chinese domestic goose of swan goose origin, performed large-scale re-sequencing of Chinese domestic geese and examined signs of selection and population structure within the Chinese domestic goose breeds. In addition, Ouyang et al. studied phylogenetic relationship of Anatidae species and suggested a conservation strategy for indigenous Chinese domestic goose breeds. The authors identified signatures of selection between white and grey Chinese domestic geese and examined further one of these identified candidate genes (EDNRB2), as this gene has an effect on coat and plumage color of several domestic animal species. The authors identified a 14-bp insertion in the EDNRB2 gene to be perfectly associated with the white plumage phenotype of Chinese domestic geese. The authors suspected an introgressed origin for the insertion region, but failed to pinpoint potential species from which this insertion could have originated. Ouyang et al. also identified selection signatures between the Xingguo gray goose

and other goose breeds, potentially conveying breed specific characteristics.

An annotated high-quality chromosome-level genome assembly of Chinese domestic goose is a welcome addition to the genomic recourses of domestic geese. The sample size of genome-resequencing was impressive. It was also noteworthy that this research focused on indigenous breeds and conservation of their genetic resources, as often research on domestic animals tends to focus on high-output commercial breeds. Also, this study made interesting findings regarding the genomic background of the white plumage phenotype. This study is valuable addition to goose domestication genetics. However, I think that the manuscript could be written more clearly, especially the Introduction section. At some parts the Introduction seemed to assume that the reader is familiar with the special characteristics of domestic geese, which might not be the case. Also, in the Result section some parts could be moved to Discussion section. The authors state that the 14-bp insertion must have introgressed from another unknown goose species, but fail to provide any tangible evidence for this. I do not think the authors can make such a definitive claim based on the evidence presented in the paper and should consider alternative explanations as well, e.g. hard selective sweep and genetic hitchhiking. More detailed comments to improve the paper are presented below line by line.

Response: We are grateful for your positive comment on this study and greatly appreciate your constructive comments contributing to the improvement of our manuscript. According to your comments, we have rewritten the Introduction section and reanalyzed the feather color phenotypes of domestic geese in the revised manuscript, see below for details.

1. Title: The title “Historical relatedness, conservation status, and signatures of selection in the Chinese indigenous goose” seems misleading as there were no analyses studying historical relatedness. To me, historical relatedness implies a temporal aspect, e.g. samples of domestic goose from museum or archaeological contexts or simulations. Please re-phrase the title to better reflect the content of the current manuscript. E.g. chromosome assembly, genome annotation and re-sequencing would much better reflect the manuscript’s content.

Response: Thank you for this valuable suggestion. Taking into account your comments and the content of revised manuscript, we have modified the manuscript title to “Chromosome-level genome and population genomics reveal evolutionary characteristics and conservation status of Chinese indigenous geese”.

2. Abstract: Line 20: In this line you mention conservation biology. Conservation biology refers to protection of endangered species and you should explain that you refer to conservation of indigenous domestic breeds and their genetic resources.

Response: Thank you for your reminder. Considering the abstract word limit, we have removed “conservation biology” and rephrased this section in lines 20–35.

3. Line 23-24: You refer to fatty liver and unique immune system of the goose. However, the sentence requires the reader to know what fatty liver and unique immune system of goose are. Replace “fatty liver” with high fat storage capability in goose liver or something similar. Also very briefly explain, what you mean by unique immune system of the goose. How it is

unique? Do you mean high disease tolerance?

Response: Thank you for your reminder. We have changed “fatty liver” to “fat storage capability in goose liver” and rephrased this section in lines 20–35.

4. Line 24-27: *This sentence does not exactly reflect the content of this paper. You have not studied the Yili breed nor all the Chinese domestic goose breeds. This information about the Yili breed is established in previous studies, but your abstract gives the impression that this breed is involved in your study. I suggest that you rephrase this sentence as “Population structure analysis verified that all studied Chinese domestic goose breeds descended from the swan goose (*Anser cygnoides*) and the studied European domestic goose breed from the greylag goose (*Anser anser*)” or similar.*

Response: Thank you for your constructive suggestion. Due to the maximum word limit for Abstracts is 150 words, we have rephrased this sentence by “Genomic resequencing of 994 geese was used to investigate the genetic relationship of geese, which supports the dual origin of geese” in revised manuscript (lines 25–26).

5. Line 31: *Change “the 14-bp” to “a 14-bp”*

Response: Thank you. We have changed “the 14-bp” to “a 14-bp” in lines 29.

6. Line 32-33: *See my later comments in the main text regarding the interspecific hybridization and rephrase the text accordingly.*

Response: Thank you. According to your later comments, we have rephrased this sentence in lines 29–33.

7. Line 34: *Change “the marker-assisted” to “a marker-assisted”*

Response: Thank you for your suggestion. We have changed “the marker-assisted” to “a marker-assisted” in line 34.

8. Main text: Line 39: *Please add reference for this.*

Response: Thank you for your reminder. According to your comments above, we have rephrased the introduction part and added related references in the revised manuscript.

9. Line 39-41: *Could you please state the evidence to which this is based on in the Bao 1996 paper. I don't have access to the original reference and it seems to be in Chinese, thus it would be beneficial for the reader if you could state to which archeological evidence this date is based on. E.g. context of the archeological finds, size of the bones etc.*

Response: Thank you for your advice. A recent article published in PNAS titled “Multiple lines of evidence of early goose domestication in a 7,000-y-old rice cultivation village in the lower Yangtze River, China” updated the domestication history of goose. We have re-described the domestication history of geese and corrected the literature in accordance with your suggestion in the revised manuscript (lines 40–42).

10. Line 41-43: *Please state what the distinctive characteristics of the goose are and what their economically important values are. The word “characteristics” is misspelled, please*

correct. Also, the goose is general term and refers to variety of species. Do you refer to all goose species or do you mean specifically domestic goose (and domestic goose generally or specifically Chinese domestic goose). As domestic geese have dual origin (swan goose and greylag goose) it is important that you establish which population you are referring to by domestic goose.

Response: Thank you for suggestion. According referees' comments, we have rephrased the Introduction section (lines 39–96). The goose here refers to the domesticated geese.

11. Line 48: Has all 30 breeds been characterized with genetic analyses? If this is not the case, change genetic to phenotypic.

Response: Thank you for reminder. We have changed “genetic” to “phenotypic” in line 44.

12. Line 58-60: Please state in more detail what the biological and economic significance of the goose means.

Response: Thank you. According to your comments above, we have rephrased the Introduction section and modified this sentence in lines 39–40.

13. Line 66: I think you need add a little bit background information why plumage color is of special interest. Now plumage color is suddenly brought up without any connection to what is previously said in the paper. Maybe it could be linked to color polymorphism segregating in domestic geese populations or early selection target in goose domestication and thus of special interest, for example.

Response: Thank you for your constructive advice. We have added the background information of plumage color in the revised manuscript (lines 72-83).

14. Line 66-69: This sentence could be understood so that you are saying that microsatellites, mtDNA or GBS-data are not reliable, only genome re-sequencing is reliable. Please rephrase. These methods are reliable, but can only be used to answer certain specific research questions or provide much more limited information. Also, I feel that you are not giving proper credit to genome-wide studies already done on Chinese domestic goose, for example:

•Deng et al. 2021 Integrative analysis of histomorphology, transcriptome and whole genome resequencing identified DIO2 gene as a crucial gene for the protuberant knob located on forehead in geese

•Li et al. 2020 Pacific Biosciences assembly with Hi-C mapping generates an improved, chromosome-level goose genome

•Liu et al. 2021 Genomic characteristics of four different geese populations in China

*•Ren et al. 2021 Pooled Sequencing Analysis of Geese (*Anser cygnoides*) Reveals Genomic Variations Associated With Feather Color*

•Wen et al. 2021 Genomic scan revealed KIT gene underlying white/gray plumage color in Chinese domestic geese

•Xi et al. 2020 A 14-bp insertion in endothelin receptor B-like (EDNRB2) is associated with white plumage in Chinese geese

•Gao et al. 2016 Genome and metagenome analyses reveal adaptive evolution of the host and interaction with the gut microbiota in the goose

•Lu et al. 2015 *The goose genome sequence leads to insights into the evolution of waterfowl and susceptibility to fatty liver*

Even though you mention many of these references later in the paper, here you give the impression that there is almost no genome-wide research on Chinese domestic goose, which is not the case. Referencing the previous studies places the results of your study in a correct context.

Response: Thank you for your comments. We do also feel that this sentence is inappropriate. According to your comments above, we have rephrased this part (lines 54–83).

15. Line 76: *Does ancestral populations mean the ancestral species the swan goose and the greylag goose?*

Response: Thank you. The ancestral populations include the swan goose and the graylag goose. According to your comments above, we have removed this sentence.

16. Line 81-83: *This sentence was not clear, application of what? Please rephrase.*

Response: Thank you. We have rephrased this sentence in the revised manuscript (lines 49–50) and hope that it is clear.

17. Line 88: *I'm a bit puzzled by the use of the word "germplasm" in here. Germplasm means: plant or animal material (such as seeds, pollen, rootstock, or sperm) that is collected and stored chiefly for future use in breeding, conservation, or research (Merriam Webster dictionary). If I understood correctly your material was not germplasm as most breeds were not involved in conservation programs. I believe investigating signs of selection would better describe your analyses.*

Response: Thank you for your valuable insight. We have modified this sentence in lines 88–90.

18. Line 91: *Change "origination" to "origin".*

Response: Thank you. We have changed "origination" to "origin" in line 91.

19. Line 95-96: *Change "the molecular mechanism of white feather formation" to "causal mutation of white plumage in Chinese domestic goose breeds"*

Response: Thank you for your valuable suggestion. We have changed "the molecular mechanism of white feather formation" to "causal mutation of white plumage in Chinese domestic goose breeds" (lines 95–96).

20. Results section: *Part of the results felt more discussion than results. There were also many references to other studies, but the Result section should be based on your results with minimal references to other studies. Please go through the Results section and move discussive parts to the Discussion section.*

Response: Thank you for your suggestion. We have carefully checked the Results section and moved discussive parts to the Discussion section as much as possible.

21. Line 102: *Supplementary Fig. S1 has poor quality and thus difficult see the method*

overview. Please provide a higher quality image.

Response: Thank you. We have changed the Supplementary Fig. S1.

22. Line 110: Fig. 1b is before Fig. 1a. Please make sure that figure panels are referenced in order of appearance, thus Fig. 1a should come first.

Response: Thank you. We have modified this figure and described in order of appearance in the main text.

23. Line 172-176: Is this based on re-sequencing data?

Response: Thank you. This is based on re-sequencing individuals used in the following population structure analyses. We have rephrased this sentence (lines 165–168) and hope that it is clarity.

24. Line 176: Method S3 is mentioned before method S2. Please change the order.

Response: Thank you. We have changed the order in the revised manuscript.

25. Line 187: Fig. 1d is before Fig. 1c.

Response: Thank you. We moved Fig. 1d to Supplementary Fig. S7 and described in order of appearance in the main text.

26. Line 237: Please include the species of the wild geese in parenthesis.

Response: Thank you for your advice. We have rephrased this sentence by “The Chinese group consisted of 772 XGG, 51 FCG, 50 GFW, 11 LHW, 50 LXW, and 5 ACy, and the European group consisted of 50 LDG and 5 AAn” (lines 220–222).

27. Line 292-294: In the previous studies which you refer, the marker type was microsatellites. Microsatellites are multiallelic and highly polymorphic while SNPs are usually biallelic. Thus, due to inherent properties of these marker types, heterozygosity in SNPs is generally lower than in microsatellites. You cannot directly compare the heterozygosity value in microsatellites and SNPs. Please re-write this sentence.

Response: Thank you for your constructive comment. We have rewritten this section in lines 265–282.

28. Line 297-235: This whole chapter felt more Discussion than Results. Consider moving most of this chapter to Discussion.

Response: Thank you for your suggestion. According to editor and other reviewers' comments, we have removed this whole chapter, which will be reflected in our other studies.

29. Line 299-301: See the previous comment. The genetic diversity in Chinese domestic goose seemed to be on same level as the wild progenitor species the swan goose, with the exception of FROH. Compared to Landaise breed, the genetic diversity of Chinese domestic geese was much higher. Please also rephrase this sentence.

Response: Thank you. We have rephrased the genetic diversity part in the above section. This whole chapter belongs to the conservation strategy study, we have removed this chapter.

30. In addition, in line 278 you say that LHW and LXW might be same breed. Thus, establishing conservation strategy for LHW alone does not seem justified.

Response: Thank you for your suggestion. We strongly agree with you. The conservation strategies should be established for LHW and LXW. We have removed this section. We will improve this section in our other studies.

31. Line 306: Abbreviation IBS appears the first time, so please write in parenthesis what this abbreviation stands for.

Response: Thank you for your advice. We have removed this section. We will improve this section in our other studies.

32. Line 326-327: I suggest re-naming this paragraph as “Selections signatures between white and grey goose populations” to better reflect the analyses performed. To me this chapter gives the impression that you searched specifically the locus for white plumage and in this case GWAS would have been more appropriate. However, you performed first a sweep analysis to search for signatures of selection and then you focused on a particular candidate gene identified by this analysis because of its association with plumage and coat color in other species, or did I misunderstood the analyses you did?

Response: Yes, we performed zFst and zHp analyses. We have renamed this chapter to “Selections signatures between Chinese white and gray geese” in line 283.

33. Line 332: Should it be “genomic signatures of selection”. Also, what do mean by shared genes? Do you mean that these genes were outliers both in zHp and zFst methods?

Response: Thank you for your constructive suggestion. We have changed “genomic signatures” to “genomic signatures of selection” in line 287. The shared genes are identified as outliers in both zHp and zFst methods. To avoid confusion, we replaced “shared genes” with “overlapping genes” in lines 288–289.

34. Line 337: Does ancestral populations mean swan and greylag goose? Are wild species and domestic geese listed in Table S26?

Response: Thank you for your question. The ancestral populations refer to the swan goose and greylag goose, wild species, and domestic geese are also listed in Supplementary Table S24.

35. Line 353: Change “nucleotide difference by more than 53” to “more than 53 nucleotide differences”.

Response: Thank you for your suggestion. We have changed “nucleotide difference by more than 53” to “more than 53 nucleotide differences” in lines 297–298.

36. Line 354-357: Could provide more information on this. E.g. how you calculated that mutational differences cannot explain the haplotype. What is the mutation rate you used? Also, you say that domestication history of the Chinese domestic geese and selective sweep events cannot explain the haplotype, so could explain this reasoning a bit more. As loss of

polymorphism around the selected locus characterizes a sweep region, so why hard sweep was ruled out as a possible explanation? Leucistic geese (white plumage) are of course very rare in nature but such geese are observed in the wild. Thus, white geese could appear in the domestic grey goose population and there would be strong artificial selection for this attractive phenotype. In domestic animal population the effective population sizes can be very low, which could lead to a rapid fixation of the sweep region.

This seems more discussion than results, please move this part to Discussion section.

Finding of such haplotype block with very low polymorphism in white geese is very interesting.

Response: Thank you for your valuable comments. The conclusion of introgressed haplotype of white geese was based on the appearance of white geese (no more than 2,000 years), mutation rate (1×10^{-8} , base pair per generation), generation interval (3 years) and selective sweep events, it is almost impossible to form such a large opposite haplotype (285 variants) in white geese during a short domestication time. However, following several of your subsequent suggestions and speculation in this comment, we carefully examined the results in this section and compared these 285 variants in ancestral populations, we found that only the 14-bp out of 285 variants in the white feather haplotypes was a derived allele, which absent in ancestral populations (ACy and AAn, $n = 65$), four closely related species (*Anatidae*, $n = 24$), rather than 285 derived variants we considered before. According to Chinese Waterfowl, gray geese were domesticated from ancestors, while white geese were artificially bred from a few gray geese after mutation⁶. We thus reasonably speculate that the 14-bp was a natural mutation that occurred during the domestication process of the gray geese, and the linkaged variants with this mutation in these near fixed white feather haplotypes can be explained genetic hitchhiking in the process of selective sweep, which consisted with your speculation. The detailed description see lines 291–317 in Results section and lines 406–434 in Discussion section.

37. Line 357-358: You cannot actually prove that the 14-bp insertion in EDNRB2 is an ancient haplotype as the ancestor of Chinese domestic geese, the swan goose, does not possess this insertion nor any of the wild relative species.

Response: Thank you. We strongly agree with you, so we have rephrased this section in the revised manuscript.

38. Line 362: Yes, it is possible that the haplotype has introgressed from an extinct species. However, as the swan goose does not possess the white feather haplotype, the introgression must have happened most probably after domestication which would place the introgression event in the last 6000 years and thus it is not very “ancient”. However, I don’t think your analyses are sufficient to try to pinpoint potential introgression events and looking into the origin of the white feather phenotype would require additional introgression analyses, which are out of the scope of this manuscript. Thus, I suggest that you are more cautious before making definitive statements or you need to provide more information on how you determined introgression as the only alternative.

Response: Thank you for your constructive insight. We strongly agree with you, so we have rephrased this section in the revised manuscript.

39. Line 368: *Overlapped by zHp and Fst methods?*

Response: Yes.

40. Line 388: *I'm uncertain to which comparative genomic analysis are you referring to. Please re-write.*

Response: Thank you. This refers to the positive selection analysis in comparative genomic analysis (lines 204–206), we have changed “comparative genomic analysis” to “positive selection analysis” in lines 344.

41. Line 396: *Related to my previous comments, “conservation biology” is out of context and you should mention conservation of genetic resources of indigenous breeds or something similar.*

Response: Thank you for your advice. We have changed “conservation biology” to “conservation of genetic resources of indigenous breeds” in line 350.

42. Line 402: *Long evolutionary process? The evolutionary history of domestic species is relatively short, so please re-phase this sentence.*

Response: Thank you for your advice. We have rephrased this sentence in line 356.

43. Line 402-406: *Can you really say that expansion of Eph/ephrin bidirectional signaling pathway in XGG had lead to sensitive and highly alert characteristics of XGG geese as all geese are alert animals. I don't think it is possible to draw such a straight line in here.*

Response: Thank you for your comment. Here, we constructed the phylogenetic tree of *Anatidae* with *Phasianidae* as the outgroup, and then explored the expanded gene families of XGG during the evolutionary process of *Anatidae* species. We here mainly highlight the adaptive evolutionary features of XGG in *Anatidae*. See lines 353–360 for detailed modifications.

44. Line 411-412: *Change “As the offspring of a migratory bird” to “As the Chinese domestic goose has descended from a migratory bird” or similar.*

Response: Thank you for your advice. We have changed “As the offspring of a migratory bird” to “As the Chinese domestic goose has descended from a migratory bird” in the line 360.

45. Line 432-433: *You say that: “then evolved the current population genetic structure”, was there in reality a lack of population structure between these varieties?*

Response: Thank you for your careful review. This may require more data to validate, so we have modified this sentence in the revised manuscript (lines 388–390).

46. Line 440-445: *Related to my previous comment on line 292-294, you cannot directly compare heterozygosity values of microsatellites and SNPs with each other due to different properties of these markers. Instead, the heterozygosity in indigenous Chinese domestic geese seems to be on rather healthy level compared to the greylag goose derivative, the Landaise*

goose. You can look also the SNP study by Heikkinen et al. 2020 (*Long-Term Reciprocal Gene Flow in Wild and Domestic Geese Reveals Complex Domestication History*) to compare heterozygosity values with different European domestic goose breeds as this study uses SNP data. Please re-write this section of your discussion. You can justify the need for conservation of indigenous goose breeds with the need to preserve genetic variation, as indigenous breeds may harbor unique genetic variation for e.g. disease tolerance or tolerance for different environmental conditions.

Response: Thank you for your constructive suggestion. We have rewritten this section in lines 390-405.

47. Line 477-479: Could you please state what good results mean? Did you observe no increase in inbreeding related values?

Response: Thank you. We have removed this content, which will be reflected in our other studies.

48. Line 492: I missed the mutation rate in the results, but was actually here. Please state the unit of the mutation rate, was it base pair per year or per generation or what. As I have previously commented, I'm a bit concerned about this statement as you cannot pinpoint the species or a ghost lineage from which this insertion could have introgressed.

Response: Thank you. We have rephrased this section in lines 406–434.

49. Line 497-500: This sentence was not clear, please re-phrase.

Response: Thank you. We have rephrased this sentence in lines 429–432.

50. Line 520: Change “involved” to “involving”

Response: Thank you. We have changed “involved” to “involving” in line 450.

51. Line 528-529: Change “Genomic” to “genomic”. Please provide reference for this or describe the extraction protocol in more detail.

Response: Thank you. We have changed “Genomic” to “genomic” and provided a URL for DNA extraction protocol (lines 461–463).

52. Line 709-715: By this analysis you are searching for signs of selection between white and grey geese, not particularly of the white plumage because you identify also other alleles selected between the white and grey geese. I think you should be more careful about the difference between selective sweep analysis and GWAS (genome-wide association analysis), which is the method to discover association between certain genomic regions and a specific trait, such as the plumage color. Please re-phrase that you searched for signs of selection between white and grey geese and then focused further analyses on one gene of interest, which has been linked to plumage or coat color in other species.

Response: Thank you for your reminder. According to your comments, we have reworded this paragraph to “Selections signatures between Chinese white and gray geese” in line 635.

53. Fig.3 legend: Line 1022: Change “anser” to “Anser”. Also, I didn't note that there was

Cygnus in the panel C. Please remove “*cygnus*” and place in the text describing panel D.

Response: Thank you for your advice. We have changed “anser” to “*Anser*” in line 977. In fact, the “Wild goose” in the panel C includes the genera *Anser* and *Cygnus* of the *Anatidae* family, we have rephrased this sentence in line 977 and hope it is clarity.

54. Table2: Line 1044: Please add to the note explanation what the breed abbreviations in the table mean.

Response: Thank you for your reminder. We have added the full names of these breed abbreviations in the lines 996–998.

1. Chen, Y.X. Chinese waterfowl (Agricultural Press 1990).
2. Borges, R. et al. Gene loss, adaptive evolution and the co-evolution of plumage coloration genes with opsins in birds. *BMC Genomics* **16**, 751 (2015).
3. Wang, Y., Li, S.M., Huang, J., Chen, S.Y. & Liu, Y.P. Mutations of TYR and MITF Genes are Associated with Plumage Colour Phenotypes in Geese. *Asian-Australas J Anim Sci* **27**, 778–783 (2014).
4. Wen, J. et al. Genomic scan revealed KIT gene underlying white/gray plumage color in Chinese domestic geese. *Anim Genet* **52**, 356–360 (2021).
5. Zhou, Z. et al. An intercross population study reveals genes associated with body size and plumage color in ducks. *Nat Commun* **9**, 2648 (2018).
6. Li, J. et al. A new duck genome reveals conserved and convergently evolved chromosome architectures of birds and mammals. *Gigascience* **10** (2021).

Reviewers' comments:

Reviewer #1 (Remarks to the Author):

This revision has improved significantly. However, there are still several issues must be resolved. As new revision introduced new important issues.

(1) page 7, line 204 to 215. This new revision misunderstands the adaptation evolution in one local breed and adaptation evolution in one species. Positively selection using the branch-site model revealed the PSG in one species. Goose liver has superior fat storage capacity should be explained as the general biology in goose. So the PSG in goose or the PSG in the XGG means the different issues, especially authors stated the PSG in the XGG in Anatidae. XGG don't show strong liver fat synthesis among goose.

(2) Page 12, line 335-340. PRLR gene were said strongly selected in XGG population. Authors should give some preliminary data to support this. And discuss this.

(3) All the references should be revised according to instruction.

(4) Many subscripts are wrongly wrote in the paper.

Reviewer #2 (Remarks to the Author):

The authors made a significant effort to review the initial manuscript and adhered to the proposed suggestions and changes of the reviewers and my own which were the most extensive.

Still, the authors have not replied to the 104th comment regarding the diversity of the SNPs in the 222 dataset. "Could the highest number of detected SNPs be due to the highest breed diversity in 222 datasets?"

The text became significantly more organized and by removing the conservation part the manuscript, become more readable and understandable.

However, I present some minor suggestions below to help improve the manuscript that should be verified in all the material.

1) Line 46: it appears "Tian fu" goose, but in Table 1 it shows as "Tianfu", which is the correct spelling?

2) Line 75: what does "down jackets" mean?

3) Line 244: replace "tree of 845 geese" with "tree of the aforementioned 845 geese"?

4) Table 2: title should mention "geese" instead?

5) There is a duplicated reference, numbers 68 and 71, should be fixed.

Reviewer #3 (Remarks to the Author):

Manuscript by Ouyang et al. was much improved from the first version. Writing about the conservation strategy of indigenous goose breeds in a separate paper makes this article more readable and less heavily packed with information. The authors had also revised their analyses and improved the readability of the article. Even though the article was mainly well written, I feel that the writing could be improved with a proofreading to check the English language. I have some minor suggestions to further improve some details of the text, these are stated below line-by-line.

Abstract, line 23: Add the species scientific name in parenthesis as well to show the origin of XGG geese, for example, ... indigenous goose (Xingguo gray goose, XGG; *Anser cygnoides*)

Abstract, line 26: Add the species in parenthesis after .."dual origin of geese". For example: "... which supports the dual origin of geese (swan and greylag goose [*A. anser*])."

Abstract, line 27: Change "rich" to "high"

Abstract, line 29: Add the full name of the gene EDNRB2 in parenthesis

Abstract, line 30: Add "Chinese" before domestic geese as European domestic goose does not appear to carry the insertion in EDNRB2 gene ...the white plumage of Chinese domestic geese...

Line 54: Add "breed" after Yili goose, ...(except for the Yili goose breed)

Line 57: Could you indicate by giving the species or genus names if the float grass refers to a single species or a group of grasses with similar growth habits

Line 73: Add "Chinese" before geese because European domestic geese do have more colours such as buff, blue and piepald

Line 72-73: Please justify why the white feather color is important economic trait, e.g. white feathers preferred in consumer products (mattresses, coats etc.) and preferred for meat production due to faster growth rate.

Line 77-79: Open the abbreviations the first time they appear in the text e.g. MITF (Melanocyte Inducing Transcription Factor) and do the same with all gene names

Line 193: There seems to be quite many XGG-specific gene families. Which is quite curious since all the four genomes compared came from the breeds of the same species so one would expect the same gene families to be present in all the breeds. Is it possible that analytical shortcomings could explain the differences e.g. differences in annotation process or genome sequencing process?

Line 204: Change "adaption" to "adaptive"

Line 249-250: Was there only specific individuals removed from these LHW and AAn populations? Thus: ... and the outlier individuals from LHW and AAn were removed. Your current sentence could be understood that you removed whole LHW and AAn populations as outliers

Line 273: Replace "richer" with "higher"

Line 293-294: Are the populations and species same as in Table S24? If so, please add reference to this table in parenthesis

Line 302-317: This part and Discussion in lines 406-434 felt repetitive. Please remove repetition, for example, by removing the parts in Results which are also told in the Discussion

Line 316-317: Or it could be from other population unstudied here? The white allele could be segregating in other gray goose breeds as well.

Line 322-323: Add "based on zFst and zHp analyses" after functional genes. E.g. We further focused on 21 overlapped functional genes based on zFst and zHp methods...

Line 359-360: I didn't quite understand this sentence. What is meant by migratory habits of geese during domestication?

Reviewer 1:

This revision has improved significantly. However, there are still several issues must be resolved. As new revision introduced new important issues.

Response: Thank you for your approval of the revised manuscript; this helps build our confidence in further improving our manuscript. According to your comments, we have revised the manuscript one by one, and hope that the revised manuscript addresses your concerns and be clear and readable.

1. page 7, line204 to 215. This new revision misunderstands the adaptation evolution in one local breed and adaptation evolution in one species. Positively selection using the branch-site model revealed the PSG in one species. Goose liver has superior fat storage capacity should be explained as the general biology in goose. So the PSG in goose or the PSG in the XGG means the different issues, especially authors stated the PSG in the XGG in Anatidae. XGG don't show strong liver fat synthesis among goose.

Response: Thank you for your careful review. We strongly agree with you that positive selection using the branch-site model revealed the PSG in one species. We have rephrased this part in the revised manuscript (lines 206–218).

2. Page 12, line 335-340. PRLR gene were said strongly selected in XGG population. Authors should give some preliminary data to support this. And discuss this.

Response: Thank you for this valuable suggestion. We have added Supplementary Table S25 in line 335 to support the selection of *PRLR* gene in the XGG population and discuss the possible effects of *PRLR* gene in lines 333–338.

3. All the references should be revised according to instruction.

Response: Thank you for your reminder. We have revised the references according to instruction of *Communications Biology*.

4. Many subscripts are wrongly wroted in the paper.

Response: Thank you for your reminder. We have carefully checked the full text and made changes to the revised manuscript.

Reviewer 2:

The authors made a significant effort to review the initial manuscript and adhered to the proposed suggestions and changes of the reviewers and my own which were the most extensive.

Still, the authors have not replied to the 104th comment regarding the diversity of the SNPs in the 222 dataset. “Could the highest number of detected SNPs be due to the highest breed diversity in 222 datasets?”

The text became significantly more organized and by removing the conservation part the manuscript, become more readable and understandable.

However, I present some minor suggestions below to help improve the manuscript that should

be verified in all the material.

Response: Thank you for your recognition of the revised manuscript and providing us a chance to revise our manuscript again. We are sorry for not replying to the previous 104th comment regarding the diversity of the SNPs in the 222 dataset. As you consider, the 222 dataset (10× depth) which includes seven breeds/species has the highest number of detected SNPs largely due to the highest breed diversity, while the aforementioned 772 dataset (1× depth) only contains one breed.

1. Line 46: it appears “Tian fu” goose, but in Table 1 it shows as “Tianfu”, which is the correct spelling?

Response: Thank you for your reminder. We have changed “Tian fu” to “Tianfu” in line 50 and have checked full text.

2. Line 75: what does “down jackets” mean?

Response: A down jacket is a coat that has been insulated with soft and warm under feathers from ducks or geese. We have rephrased this sentence in lines 79–81.

3. Line 244: replace “tree of 845 geese” with “tree of the aforementioned 845 geese”?

Response: Thank you for this valuable suggestion. We have replaced “tree of 845 geese” with “tree of the aforementioned 845 geese” in line 245.

4. Table 2: title should mention “geese” instead?

Response: Thank you for your suggestion. We have replaced “goose” with “geese” in the title of Table 2.

5. There is a duplicated reference, numbers 68 and 71, should be fixed.

Response: Thank you for your reminder. We have removed the redundant reference in the revised manuscript.

#Reviewer 3:

Manuscript by Ouyang et al. was much improved from the first version. Writing about the conservation strategy of indigenous goose breeds in a separate paper makes this article more readable and less heavily packed with information. The authors had also revised their analyses and improved the readability of the article. Even though the article was mainly well written, I feel that the writing could be improved with a proofreading to check the English language. I have some minor suggestions to further improve some details of the text, these are stated below line-by-line.

Response: We are grateful for your positive comment on this study. We have carefully checked the full text and made some detailed revisions according to your comments. For grammar issues, we have made professional grammatical revisions to the manuscript through professional organization (<http://www.letpub.com>).

Certificate of English Language Editing

Manuscript Title:

Historical relatedness, conservation status, and signatures of selection for white feathers in the Chinese indigenous goose

Date of Revision:

July 26, 2022

Abstract:

Geese are herbivorous birds that play an essential role in the agricultural economy. To the best of our knowledge, the present study was the first to construct the chromosome-level genome of a Chinese indigenous goose (the Xingguo gray goose, XGG; *Anser cygnoides*) and analyze the adaptation of fat storage capacity in the goose liver during the evolution of Anatidae. Genomic resequencing of 994 geese was used to investigate the genetic relationships of geese; the results supported the dual origin of geese (*Anser cygnoides* and *Anser anser*). Chinese indigenous geese showed higher genetic diversity than European geese, and a scientific conservation program could be established to preserve genetic variation for each breed. We also found that a 14-bp insertion in endothelin receptor B subtype 2 (EDNRB2) that determines the white plumage of Chinese domestic geese was a natural mutation, and the linkage of alleles rapidly increased in frequency as a result of genetic hitchhiking, leading to the formation of completely different haplotypes of white geese under strong artificial selection. These genomic...

This document certifies that the manuscript listed above was copy edited for English language by LetPub, with regard to grammar, punctuation, spelling, and clarity. All of our language editors are native English speakers with long-term experience in editing scientific and technical manuscripts. We are committed to leveling the playing field for researchers whose native language is not English.

- Documents receiving this certification should be regarded as having undergone professional editorial revision for English language before submission. However, the authors may accept or reject LetPub's suggestions and changes at their own discretion and LetPub does not have editorial control over the submitted documents.
- The language quality of the submitted document is the sole responsibility of the submitting authors subject to those authors' adherence to LetPub's revisions and instruction. LetPub's provision of service does not constitute a guarantee or endorsement of the authors' work herein.
- Neither the research content nor the authors' intended meaning were altered in any way during the editing process.
- If you have any questions or concerns about this edited document, please contact us at support@letpub.com

LetPub is an author service brand owned and operated by Accdon LLC. Headquartered in the Boston area, we are a full-spectrum author services company with a large team of US-based certified language and scientific editors, ISO 17001 accredited translators, and professional scientific illustrators and animators. We advocate ethical publication practices and are an official member of the Committee on Publication Ethics (COPE).

For more information about our company, services, and partnership programs, please visit www.letpub.com.

© 2022 Accdon, LLC. All Rights Reserved. Tel: 1-781-202-9968 Email: info@accdon.com Address: 400 Fifth Ave, Suite 530, Waltham, MA 02451, United States

1. Abstract, line 23: Add the species scientific name in parenthesis as well to show the origin of XGG geese, for example, ... indigenous goose (*Xingguo gray goose, XGG; Anser cygnoides*)

Response: Thank you for this valuable suggestion. We have changed “(Xingguo gray goose, XGG)” to “(Xingguo gray goose, XGG; *Anser Cygnoides*)” in line 25.

2. *Abstract, line 26: Add the species in parenthesis after .."dual origin of geese". For example: "... which supports the dual origin of geese (swan and greylag goose [A. anser])."*

Response: Thank you for your reminder. We have added the species scientific name in parenthesis (lines 28–29).

3. *Abstract, line 27: Change "rich" to "high"*

Response: Thank you for your reminder. We have changed "rich" to "high" in line 29.

4. *Abstract, line 29: Add the full name of the gene EDNRB2 in parenthesis*

Response: Thank you for your reminder. We have replaced "EDNRB2" with "EDNRB2 (Endothelin Receptor B Subtype 2)" in line 32.

5. *Abstract, line 30: Add "Chinese" before domestic geese as European domestic goose does not appear to carry the insertion in EDNRB2 gene ...the white plumage of Chinese domestic geese...*

Response: Thank you for this valuable suggestion. We have added the word "Chinese" before domestic geese in line 33.

6. *Line 54: Add "breed" after Yili goose, ...(except for the Yili goose breed)*

Response: Thank you for your suggestion. We have added the word "breed" after Yili goose in line 59.

7. *Line 57: Could you indicate by giving the species or genus names if the float grass refers to a single species or a group of grasses with similar growth habits*

Response: Thank you for your reminder. The float grass is herbs that can grow in water, here mainly referring to the *Cyperaceae*. We have added "*Cyperaceae*" in parenthesis after float grass in line 63.

8. *Line 73: Add "Chinese" before geese because European domestic geese do have more colours such as buff, blue and piepald*

Response: Thank you for your constructive advice. We have added "Chinese" before geese in line 81.

9. *Line 72-73: Please justify why the white feather color is important economic trait, e.g. white feathers preferred in consumer products (mattresses, coats etc.) and preferred for meat production due to faster growth rate.*

Response: Thank you for your valuable suggestion. We have added the sentence "White feathers are preferred in consumer products (e.g., mattresses and coats) and are preferred for meat production due to the faster growth rate of birds with white plumage." in lines 79–81.

10. *Line 77-79: Open the abbreviations the first time they appear in the text e.g. MITF (Melanocyte Inducing Transcription Factor) and do the same with all gene names*

Response: Thank you for your reminder. We have carefully checked the main text and added

full gene names when the abbreviations first appeared in the manuscript.

11. Line 193: *There seems to be quite many XGG-specific gene families. Which is quite curious since all the four genomes compared came from the breeds of the same species so one would expect the same gene families to be present in all the breeds. Is it possible that analytical shortcomings could explain the differences e.g. differences in annotation process or genome sequencing process?*

Response: Thank you for your question. XGG has 11,648 gene families, only 15 XGG-specific gene families, and the rest are shared with other domestic geese (lines 195–205 and Supplementary Fig. S6).

12. Line 204: *Change “adaption“ to “adaptive”*

Response: Thank you for your reminder. We have rephrased this sentence in lines 206–207.

13. Line 249-250: *Was there only specific individuals removed from these LHW and AAn populations? Thus: ... and the outlier individuals from LHW and AAn were removed. Your current sentence could be understood that you removed whole LHW and AAn populations as outliers.*

Response: Thank you for your comments. We just removed a LHW and an AAn individual deviated from the population branch. For clarity, we have changed “the outliers LHW and AAn were removed” to “the outlier individuals from LHW and AAn were removed” (line 251).

14. Line 273: *Replace “richer” with “higher”*

Response: Thanks. Replaced (line 276).

15. Line 293-294: *Are the populations and species same as in Table S24? If so, please add reference to this table in parenthesis.*

Response: Thank you for your advice. We have reordered the tables so that the populations are the same as in Table S23, we have added a reference to this table in parenthesis (line 297).

16. Line 302-317: *This part and Discussion in lines 406-434 felt repetitive. Please remove repetition, for example, by removing the parts in Results which are also told in the Discussion.*

Response: Thanks for your suggestion. We have modified the Results and corresponding Discussion of this part in the revised manuscript.

17. Line 316-317: *Or it could be from other population unstudied here? The white allele could be segregating in other gray goose breeds as well.*

Response: Thank you for your valuable insight. We strongly agree with you and have added “or other population not studied here” in line 314.

18. Line 322-323: *Add “based on zFst and zHp analyses” after functional genes. E.g. We further focused on 21 overlapped functional genes based on zFst and zHp methods...*

Response: Thank you for your advice. We have added “based on zFst and zHp analyses”

after functional genes in line 304.

19. Line 359-360: I didn't quite understand this sentence. What is meant by migratory habits of geese during domestication?

Response: Thank you for your comments. We have reworded this sentence in lines 360–362.

REVIEWERS' COMMENTS:

Reviewer #1 (Remarks to the Author):

This revision has solve the general issues mentioned.

Reviewer #2 (Remarks to the Author):

I believe the manuscript is reaching a good status for publication. However, I leave only a few remarks below.

Regarding the previous question raised concerning the 222 datasets and the number of SNPs detected, should the authors given justification be also provided in main text?

Line 24: construct?

Line 38: superfluous "a"?

Lines: 195-197: Following the comments from the fellow reviewer #3 and the authors respective answer, it seems that the word "shared" is (still) missing here. If I understand correctly, the authors reply that the 11,648 are shared among other domestic geese (the four mentioned in text?), but still this is not very clear, since only XGG is described or associated. I suggest this whole paragraph should be rephrased.

Line 265: What is the "CV-error"? First time seen abbreviation?

Line 398: Throughout the main text both terms appear "goose breeds" and "geese breeds", which one is correct? Are they both correct?

Reviewer 1:

This revision has solve the general issues mentioned.

Response: Thank you for your recognition of this work and for your very constructive comments on the improvement of our manuscript.

Reviewer 2:

I believe the manuscript is reaching a good status for publication.

Response: Thank you for your recognition of the revised manuscript and for your contribution to this article.

1. Regarding the previous question raised concerning the 222 datasets and the number of SNPs detected, should the authors given justification be also provided in main text?

Response: Thank you for your suggestion. We have modified the sentence “These sequencing reads were aligned with the reference genome XGG assembled above, 772 XGG (1×) yielded 12,415,004 SNPs, while 222 geese (10×) yielded 13,008,900 SNPs that were more abundant than XGG population, largely due to breed diversity and higher sequencing depth (Supplementary Method 4).” in the revised manuscript (lines 224–228).

2.Line 24: construct?

Response: Thank you for your correct. We have changed “construst” with “construct” in the line 23.

3.Line 38: superfluous “a”?

Response: Thank you for your carefully review. We have removed the word “the” in the line 37.

4.Lines: 195-197: Following the comments from the fellow reviewer #3 and the authors respective answer, it seems that the word “shared” is (still) missing here. If I understand correctly, the authors reply that the 11,648 are shared among other domestic geese (the four mentioned in text?), but still this is not very clear, since only XGG is described or associated. I suggest this whole paragraph should be rephrased.

Response: Thanks for pointing out. It’s a pity we didn’t describe it clearly here. In fact, we detected a total of 11,733 gene families in the 4 geese genomes, of which 9,390 gene families were shared by the 4 geese. In XGG, we detected 11,648 gene families, including 9,390 gene families shared by 4 geese, 1,850 gene families shared by 3 geese, 393 gene families shared by 2 geese, and 15 gene families unique to XGG (Supplementary Figure 6). Here we mainly describe the evolutionary properties of XGG, so 15 unique gene families were selected for enrichment analysis. The revised sentence “A comparison of four geese genomes (XGG, SCW, ZDW, and TFG) showed that a total of 11,733 gene families, while 9,390 gene families were shared. We detected 11,648 gene families in XGG, of which 15 XGG-specific gene families included 38 genes (Supplementary Figure 6).” see lines 192–195.

5.Line 265: What is the “CV-error”? First time seen abbreviation?

Response: Thanks for pointing out. We have changed “CV-error” to “cross-validation error”

in the line 264.

6.Line 398: Throughout the main text both terms appear “goose breeds” and “geese breeds”, which one is correct? Are they both correct?

Response: Thanks for pointing out. We have carefully checked the full text and changed “goose breeds” to “geese breeds” in the revised manuscript.